*Atmos. Chem. Phys.* manuscript:

# Atmospheric Measurements at Mt. Tai - Part I: HONO Formation and Its Role in the Oxidizing Capacity of the Upper Boundary Layer

Chaoyang Xue[1, 2*], Can Ye[1, 9], Jörg Kleffmann[3], Chenglong Zhang[1, 4], Valéry Catoire[2], Fengxia Bao[5], Abdelwahid Mellouki[6, 7], Likun Xue[7], Jianmin Chen[8], Keding Lu[9], Yong Zhao[10], Hengde Liu[10], Zhaoxin Guo[10], Yujing Mu[1, 4*]

[1] Research Centre for Eco-Environmental Sciences, Chinese Academy of Sciences, Beijing 100085, China

[2] Laboratoire de Physique et Chimie de l'Environnement et de l'Espace (LPC2E), CNRS–Université Orléans–CNES, Cedex

2, Orléans 45071, France

[3] Physical and Theoretical Chemistry, University of Wuppertal, Gaußstrasse 20, Wuppertal 42119, Germany

[4] Centre for Excellence in Regional Atmospheric Environment, Institute of Urban Environment, Chinese Academy of Sciences, Xiamen 361021, China

[5] Multiphase Chemistry Department, Max Planck Institute for Chemistry, Mainz 55128, Germany

[6] Institut de Combustion Aérothermique, Réactivité et Environnement, Centre National de la Recherche Scientifique (ICARE-CNRS), Cedex 2, Orléans 45071, France

[7] Environmental Research Institute, Shandong University, Qingdao, Shandong 266237, China

[8] Shanghai Key Laboratory of Atmospheric Particle Pollution and Prevention, Department of Environmental Science and Engineering, Institute of Atmospheric Sciences, Fudan University, Shanghai 200438, China

[9] State Key Joint Laboratory of Environment Simulation and Pollution Control, College of Environmental Sciences and Engineering, Peking University, Beijing, 100871, China

[10] Taishan National Reference Climatological Station, Tai'an, Shandong, 271000, China

*Correspondence to*:

Chaoyang Xue (chaoyang.xue@cnrs-orleans.fr; 86chaoyang.xue@gmail.com)

Yujing Mu (yjmu@rcees.ac.cn)

**Abstract**

A comprehensive field campaign, with measurements of HONO and related parameters, was conducted in summer 2018 at the foot (150 m a.s.l.) and the summit (1534 m a.s.l.) of Mt. Tai (Shandong province, China). At the summit station, high HONO mixing ratios were observed during this campaign (mean ± 1σ: 133 ± 106 pptv, maximum: 880 pptv), with a diurnal noontime peak (mean ± 1σ: 133 ± 72 pptv at 12:30 local time). Constraints on the kinetics of aerosol-derived HONO sources ($NO_2$ uptake on the aerosol surface and particulate nitrate photolysis) were performed and discussed, which enables a better understanding of the interaction of HONO and aerosols, especially in the polluted North China Plain. Various shreds of evidence of air mass transport from the ground to the summit levels were provided. Furthermore, daytime HONO formation from different paths and its role in radical production were quantified and discussed.

We found that the homogeneous reaction NO + OH could only explain 8.0% of the daytime HONO formation, resulting in strong unknown sources ($P_{un}$). Campaigned-averaged $P_{un}$ was about 290 ± 280 pptv $h^{-1}$ with a maximum of about 1800 pptv $h^{-1}$. Aerosol-derived HONO formation mechanisms were not the major sources of $P_{un}$. Their contributions to daytime HONO formation varied from negligible to moderate (similar to NO + OH), depending on the used chemical kinetics. Coupled with sensitivity tests on the used kinetics, the $NO_2$ uptake on the aerosol surface and particulate nitrate photolysis contributed 1.5 – 19% and 0.6 – 9.6% of the observed $P_{un}$, respectively. Based on synchronous measurements at the foot and the summit stations, a bunch of field evidence was proposed to support that the remaining majority (70 – 98%) of $P_{un}$ was dominated by the rapid vertical transport from the ground to the summit levels and heterogeneous formation on the ground surfaces during the transport. HONO photolysis at the summit level initialized daytime photochemistry and still represented an essential OH source in the daytime, with a contribution of about one-fourth of $O_3$. We provided evidence that ground-derived HONO played a significant role in the oxidizing capacity of the upper boundary layer through the enhanced vertical air mass exchange driven by mountain winds. The follow-up impacts should be considered in the regional chemistry-transport models.

# 1 Introduction

In the past two decades, atmospheric nitrous acid (HONO) has attracted numerous laboratory experiments and field campaigns because of its significant contribution to the atmospheric concentration of hydroxyl radicals (OH) and the incomplete understanding of its sources (Kleffmann, 2007). Besides the homogeneous reaction of NO with OH, various HONO formation pathways were proposed, including: a) emission from combustion processes, e.g., vehicle exhaust, domestic conbustion and biomass burning (Klosterköther et al., 2021; Kramer et al., 2020; Kurtenbach et al., 2001; Liu et al., 2017; Peng et al., 2020; Theys et al., 2020); b) heterogeneous dark and photosensitized reactions of $NO_2$ on surfaces such as soot (Ammann et al., 1998; Monge et al., 2010), organic compounds (George et al., 2005; Han et al., 2017; Stemmler et al., 2006, 2007), acids (Kleffmann et al., 1998), urban grime (Liu et al., 2019a), MgO (Ma et al., 2017), mineral dust (Ndour et al., 2008), vegetation leaves (Marion et al., 2021), etc.; c) photolytic reactions of total nitrate (particulate nitrate and adsorbed nitric acid) (Bao et al., 2018; Laufs and Kleffmann, 2016; Ye et al., 2016; Zhou et al., 2003, 2011) and ortho-nitrophenols (Bejan et al., 2006); d) emissions

from soil (Donaldson et al., 2014; Oswald et al., 2013; Su et al., 2011; Xue et al., 2019a), etc. Even though many potential HONO sources have been identified in the past, there is still a significant gap between model results and observations (Fu et al., 2019; Liu et al., 2017; Xue et al., 2020; Zhang et al., 2019a, 2019b). One of the critical puzzles is the quantity of HONO formation from the aerosol-derived sources, particularly $NO_2$ uptake on aerosol surfaces and aerosol nitrate photolysis in high-aerosol regions such as the North China Plain (NCP).

The $NO_2$ uptake on aerosol surfaces was proposed to be much less important than that on ground surfaces in previous studies because of the low S/V (surface to volume ratio) of particles compared to ground surfaces and the similar reaction kinetics on the same types of surfaces (Nie et al., 2015; Stemmler et al., 2007). However, the contribution of $NO_2$ uptake on aerosol surfaces to HONO formation in the extremely polluted region is not well constrained. For example, previous studies using box models or regional transport chemistry models found the $NO_2$ uptake on aerosol surfaces lead to a negligible impact on daytime HONO formation in the polluted NCP (Liu et al., 2019b; Xue et al., 2020; Zhang et al., 2019a, 2019b). Nevertheless, a recent chamber study (Ge et al., 2019) found a high dark $NO_2$ uptake coefficient ($2.0 \times 10^{-5}$ to $1.7 \times 10^{-4}$) on NaCl particles under high RH (90%), $NH_3$ (50-2000 ppbv), and $SO_2$ (600 ppbv) conditions. First, such severe pollution rarely occurred. Second, if such a high $NO_2$ coefficient on the aerosol surface was applied in night-time HONO budget analysis, the dominant role of $NO_2$ uptake on the ground surface in night-time HONO formation, which was already generally accepted, might be challenged (Kleffmann, 2007; Kurtenbach et al., 2001; Stutz et al., 2002; Xue et al., 2020). Besides, recent nocturnal vertical measurements of HONO in Beijing found both ground-based and aerosol-derived sources may play important roles in HONO formation during the clean period and haze period, respectively (Meng et al., 2020). Therefore, the contribution of $NO_2$ uptake on the aerosol surface to HONO formation still needs more field constraints.

The photolysis of particulate nitrate ($pNO_3$) was found to be an important HONO source in low $NO_x$ areas such as forest canopy and marine boundary layer. High enhancement factors (EF = $J(pNO_3)/J(HNO_3)$), within the range of tens to thousands, were proposed in forest areas, the marine boundary layer, and polluted areas like the NCP (Bao et al., 2020; Ye et al., 2016, 2017; Zhou et al., 2007, 2011). However, model studies with field constraints (Romer et al., 2018; Xue et al., 2020) found that the EF was moderate (7-30) rather than tens to thousands obtained in laboratory studies (Bao et al., 2020; Ye et al., 2016, 2017; Zhou et al., 2007). Moreover, a recent laboratory flow tube study (Wang et al., 2021) revealed that the EF was lower than 1 in the aqueous phase. Another flow tube study (Laufs and Kleffmann, 2016) also reported a slow HONO formation from secondary heterogeneous reactions of $NO_2$ produced during $HNO_3$ photolysis. Besides, a very recent chamber study (Shi et al., 2021) found that the EF values of airborne nitrate were lower than 10 (generally around 1), which also indicates an insignificant contribution of nitrate photolysis to HONO formation. Furthermore, when considering the large variation of EF values (from digits to thousands) in the model, model performance on HONO simulations could be improved but accompanied by large uncertainties (Fu et al., 2019; Liu et al., 2019b; Zhang et al., 2021). Therefore, HONO formation from nitrate photolysis still needs more field constraints.

In addition, the role of HONO photolysis in the oxidizing capacity of the upper boundary layer remains unclear. As there exists a significant gradient in HONO distribution, HONO photolysis was accounted to be much less important compared to $O_3$

photolysis in the upper troposphere compared to the lower troposphere (Ye et al., 2018; Zhang et al., 2009). However, in mountainous regions, mountain winds, including mountain breeze (downslope) and valley breeze (upslope) can accelerate the air mass exchange between the mountain top and the ground levels, which may affect HONO levels and the atmospheric oxidizing capacity at the summit level (Jiang et al., 2020; Schmid et al., 2020; Ye et al., 1987).

Herein, atmospheric measurements at the foot (~150 m a.s.l.) and the summit (~1534 m a.s.l.) of Mt. Tai in summer 2018 are presented in this study. Comprehensive measurements allow us to understand more about 1) the transport of ground-formed HONO and its role in the upper boundary layer; 2) HONO formation from the aerosol-derived sources as the ground-derived sources might be less effective compared to measurements near to ground surface; 3) the oxidizing capacity of the upper boundary layer and its contributors.

## 2 Experimental

### 2.1 Site Description

HONO was alternately measured at two locations: the foot and the summit of Mt. Tai (Figure 1 and S1). The foot station was inside Shandong College of Electric Power, a typical urban site (36.18°N, 117.11°E). HONO, VOCs, OVOCs, CO, $O_3$, $SO_2$, $NO_x$, $PM_{2.5}$, $PM_{10}$, $J(NO_2)$, and meteorological parameters were continuously measured at this station. Details about the foot station and the used instruments can be found in the companion study (Xue et al., 2022). The summit station (36.23°N, 117.11°E) is located inside a meteorological observatory at the eastern part of the summit of Mt. Tai, with an altitude of about 1534 m a.s.l. It is in the north part of Tai'an city (altitude: ~150 m, population: ~5.6 million), and about 60 km south of Jinan city (the capital city of Shandong province, altitude: ~20 m, population: ~8.7 million).

Since Mt. Tai is a famous tourist place, most of the tourist activities on the summit happen around the Southern Heavenly Gate, the Bixia Temple, and the Jade Emperor Peak. The most crowded period is around sunrise when visitors come for the view of sunrise. The Southern Heavenly Gate is about 1 km west of and about 100 m lower than our station. There are several small restaurants nearby, but they don't cause significant emissions as they only use electricity for the energy supply. The Bixia Temple is about 200 m west to and about 50 m lower than our station, and small anthropogenic emissions may be produced here because of the incense burning, but the impact on our measurements is expected to be negligible as a result of the fast dilution process at the summit level. The Jade Emperor Peak is about 200 m northwest of and has a similar altitude to our station. Visitors generally stay there for a short time and don't have activities that may produce significant emissions. A detailed discussion about the influence of anthropogenic emissions at the summit level on our measurement is presented in Section 3.2.1.

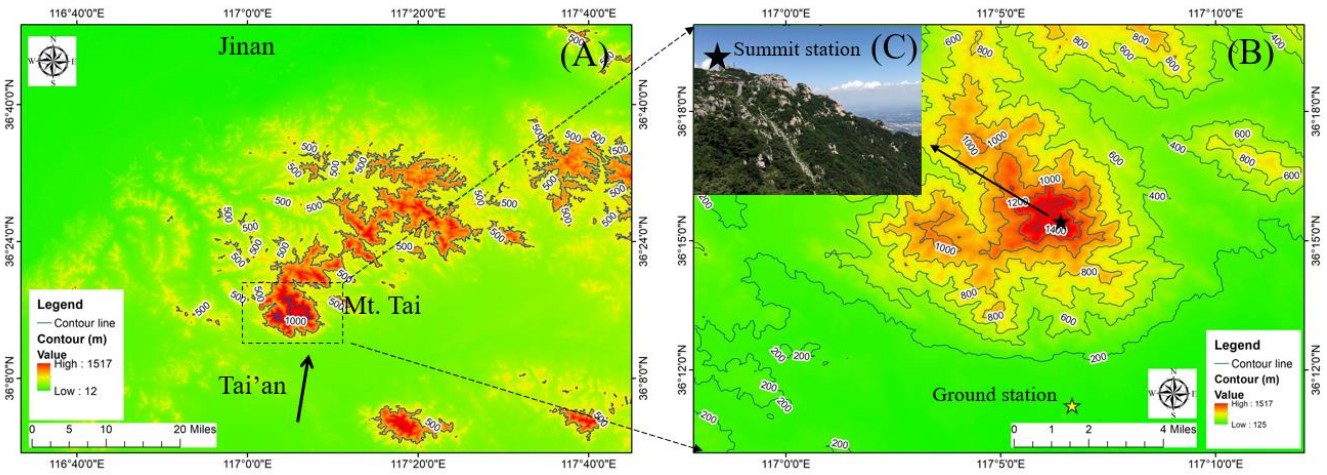

**Figure 1: Locations of Mt. Tai, the summit station, and the ground station. (A): Locations of Mt. Tai and nearby cities (Tai'an and Jinan) colored by altitude; the black arrow represents the dominated wind direction. (B): Contour map of Mt. Tai; The black and the yellow stars represent the locations of the summit and the ground stations, respectively. (C): A view of the station from the southwest. Map data were taken from the National Catalogue Service for Geographic Information (https://www.webmap.cn). Photo copyright: Chaoyang Xue.**

## 2.2 Instrumentation

During the campaign, HONO was continuously measured by the LOPAP technique (LOng Path Absorption Photometer, Model-03, QUMA GmbH, Germany) with a detection limit of 1.5 pptv for 5 min average (Heland et al., 2001; Kleffmann et al., 2006). The performance of LOPAP was well assessed and recorded in different environmental conditions (Heland et al., 2001), including low-$NO_x$ and high-altitude sites (Kleffmann and Wiesen, 2008). Besides, this LOPAP has been successfully used in our previous studies (Xue et al., 2019b, 2020). The LOPAP instrument was installed at the foot station from 29th May to 8th July 2017, and then transported to the summit station with successful measurements from 9th to 31st July 2017. At the summit station, a temperature-controlled measurement container was used to house all the instruments. The external sampling unit of LOPAP was installed on the top of the container, about 2.5 m above the ground surface. Zero air (ultrapure $N_2$) measurements were conducted 2 or 3 times per day. Liquid calibration with diluted standard nitrite solution (Sigma-Aldrich) was conducted every week. Both zero air measurements and liquid calibration were conducted after changing any solution, cleaning the instrument, or replacing any component of the instrument (the air pump was broken on 21st July and replaced by a new one on 25th July). The precision of the instrument determined from 2σ noise of the calibration was 1%. An accuracy of 7% was determined by error propagation including all known uncertainties, i.e., the concentration of the calibration standard (±3-4%) and the liquid (±1%) and gas flow (±2%) rates. Known artificial HONO formation on inlet surfaces (e.g., Zhou et al., 2002) were minimized by using the external sampling unit, with only a 3 cm sunlight-shielded glass inlet to the ambient atmosphere. Other interferences were considered of minor importance, as they were corrected for by the two-channel concept of the instrument. In addition, excellent agreement between LOPAP and DOAS techniques was observed under complex conditions in a smog chamber and in the ambient atmosphere (Kleffmann et al., 2006). $NO_2$ was measured by a Model-T500U-

CAPS-NO$_2$-analyzer (Teledyne API, USA) that utilizes a patented Cavity Attenuated Phase Shift (CAPS) technique to measure NO$_2$ in the air directly. NO and NO$_y$ were measured by API-T200U-NO$_y$-analyzer (Teledyne API, USA) based on the chemiluminescence principle coupled with a remote NO$_y$ converter via umbilical to allow measurements with a lower detectable limit of 50 pptv. PM$_{2.5}$ was measured by a SHARP 5030 monitor (Thermo Scientific, USA). CO and SO$_2$ were measured by a T300U-CO monitor (Teledyne API, USA) and a Model 43C SO$_2$ monitor (Thermo Scientific, USA), respectively. J(NO$_2$) was measured by a 4-$\pi$ J(NO$_2$) filter radiometer (Metcon GmbH, Germany). Other J-values used in this study, including J(HONO), J(O($^1$D)), and J(HNO$_3$), are calculated by the trigonometric SZA function (MCM default photolysis frequency calculation, see the companion paper and Jenkin et al. (1997)) and scaled by the measured J(NO$_2$). For instance, J(HONO) = J(HONO)$_{model}$ $\times$ J(NO$_2$)$_{measured}$ / J(NO$_2$)$_{model}$.

Water-soluble ions, including particulate nitrate (pNO$_3$) of PM$_{2.5}$, were collected by filter method and analyzed by an ion chromatograph (Liu et al., 2020) every 2 hours late June and early July, but it suffered a sampling problem after 12$^{th}$ July. Aerosol size (13.6 – 763.5 nm) distribution was measured by a Scanning Mobility Particle Sizer (SMPS, Model 3938, TSI Inc., USA) equipped with a Differential Mobility Analyzer (DMA, Model 3082, TSI Inc., USA) and a Condensation Particle Counter (CPC, Model 3775, TSI Inc., USA). Meteorological parameters (temperature, relative humidity, wind speed, wind direction) were measured by instruments from the Shandong Taishan Meteorological Station simultaneously, and details can be found in previous studies at this station (Jiang et al., 2020). In this study, 10-min averaged data were used for the following analysis. Details about the instrumentation at the foot station could be found in the companion ACP paper. Measurements at the foot station ended on 16$^{th}$ July. To compare pollutants between the foot and the summit levels during the same period (Section 3.2.2), measurements (only hourly CO, NO$_2$, PM$_{2.5}$, PM$_{10}$, O$_3$, and SO$_2$ were available) from the monitoring station (~200 m east to the foot station) were used.

## 3 Results and Discussion

### 3.1 Overview of the Observations

Figure 2 shows the meteorological parameters measured at the summit of Mt. Tai during the campaign. The air temperature (T in °C) was slightly lower (~17 °C) in the first two days compared to the period after 10$^{th}$ July (~20 °C). As clouds were frequently formed at the summit (Li et al., 2020), the observed relative humidity (RH) commonly reached 100%, with a mean of 96%. Based on the wind measurements, air mass at the summit mainly came from the south (direction of Tai'an city), with a mean wind speed (WS) of 5.1 m s$^{-1}$. In particular, during the period of 23$^{rd}$ to 26$^{th}$ July, high wind speed (1-min max: 19.5 m s$^{-1}$, 10-min max: 18.5 m s$^{-1}$) was observed, accompanied by a relatively low temperature, low pressure (p), low radiation (J(NO$_2$)), and high RH.

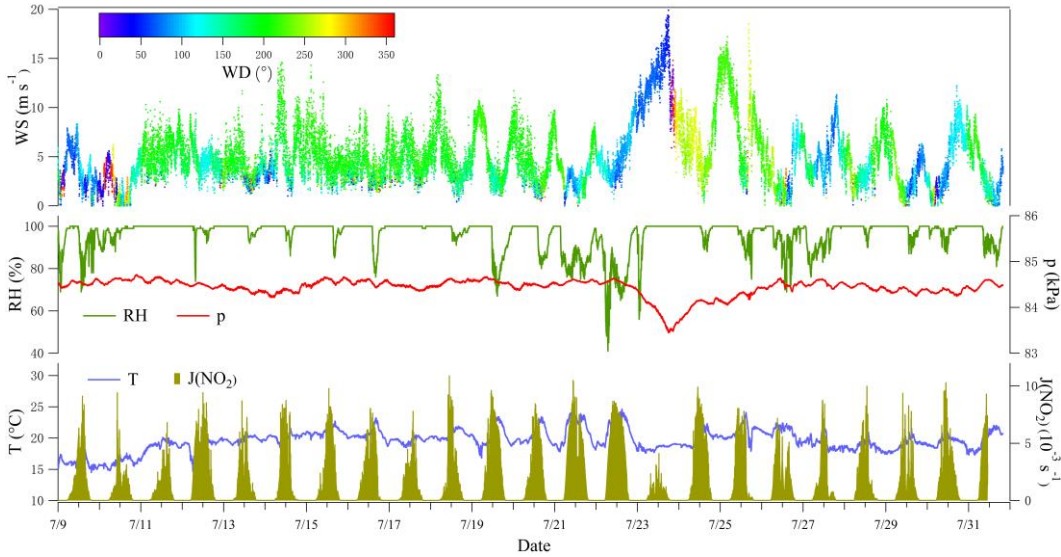

**Figure 2: Meteorological parameters measured at the summit of Mt. Tai during the campaign.**

Figure 3 illustrates the time series of HONO and related pollutants measured during the campaign. Several pollution events were observed. For example, the measured daytime $PM_{2.5}$ was generally larger than 20 µg m$^{-3}$, and high $SO_2$ mixing ratios (>1 ppbv) were observed during the daytime on some days (i.e., from 14$^{th}$ to 26$^{th}$ July). NO mixing ratios were generally lower

than 0.5 ppbv due to significant suppression by high $O_3$ levels of usually higher than 50 ppbv. $NO_2$ was generally lower than 2 ppbv with several events, during which $NO_2$ was relatively higher. Besides, the measured HONO mixing ratio varies from 1.1 pptv (close to the detection limit) to 880 pptv, with a mean of 133 pptv and a median of 101 pptv, respectively (Table 1). For the same sampling site at the summit of Mt. Tai, as listed in Table 2, the observed mean HONO mixing ratios in summer is similar to those observed at the same site in winter (150 pptv, December 2017) and spring (130 pptv, March – April 2018)

reported by Jiang et al. (2020), but the variation of HONO mixing ratios in summer was within a much narrower range (1 – 880 pptv) than in winter (0 – 1140 pptv) and spring (0.5 – 3230 pptv). With an exception for relatively lower HONO levels at altitudes higher than 2000 m or in the free troposphere (Ye et al., 2018), HONO mixing ratios are significantly higher at the summit of Mt. Tai than at other mountain sites (Table 2). For example, mean HONO mixing ratios observed at Mt. Whiteface in the USA (Zhou et al., 2007) and Mt. Hohenpeissenberg in Germany (Acker et al., 2006) were 46 and 100 (daytime)/30

(night-time) pptv, respectively. This phenomenon could be explained by fewer human activities around these mountains, while Mt. Tai locates in the middle of the NCP with a relatively high pollution level.

Note that high HONO mixing ratios were observed during the periods from 14$^{th}$ to 26$^{th}$ July, with the co-occurrence of high $SO_2$ (a primary pollutant generally emitted at the ground level with a relatively short lifetime). To better understand HONO formation and its role at different pollution levels, data were classified into two periods: high HONO period (HP, 14$^{th}$ to 26$^{th}$)

and low HONO period (LP) that covers all the other days. Statistics of observations during the two periods are summarized in

Table 1. Average HONO, NO$_y$, SO$_2$, and PM$_{2.5}$ during LP are 76 pptv, 4.7 ppbv, 0.3 ppbv, and 12 µg m$^{-3}$, respectively, slightly lower than those during HP (194 pptv, 7.0 ppbv, 0.8 ppbv, and 17 µg m$^{-3}$, respectively).

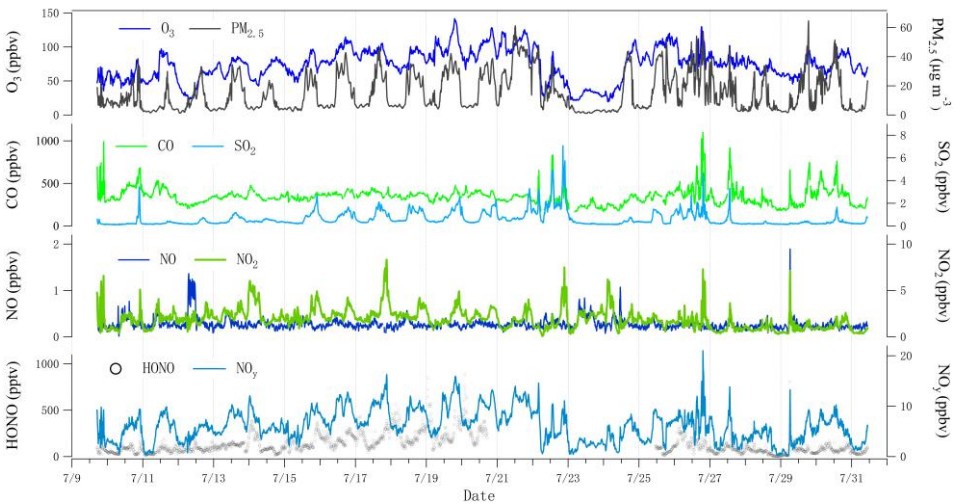

**Figure 3: HONO and related species measured at the summit of Mt. Tai during the campaign.**

**Table 1: Statistics of observations from 9$^{th}$ to 31$^{st}$ July 2018 at the summit of Mt. Tai. High HONO period (HP): 14$^{th}$ – 26$^{th}$ July; Low HONO period (LP): 9$^{th}$ – 13$^{th}$ and 27$^{th}$ – 31$^{st}$ July.**

| Parameters | Whole campaign | | | | LP/HP | | | |
|---|---|---|---|---|---|---|---|---|
| | Min | Max | Mean | Median | Min | Max | Mean | Median |
| HONO (pptv) | 1.1* | 880 | 133 | 101 | 1.1*/14 | 806/880 | 76/194 | 71/172 |
| NO (ppbv) | 0.01* | 1.89 | 0.27 | 0.25 | 0.01*/0.07 | 1.9/1.1 | 0.27/0.27 | 0.24/0.26 |
| NO$_2$ (ppbv) | 0.1* | 8.3 | 1.9 | 1.8 | 0.3/0.1* | 7.1/8.3 | 1.4/2.2 | 1.2/2.0 |
| NO$_y$ (ppbv) | 0.2 | 21.0 | 6.1 | 6.0 | 0.2/0.4 | 13.8/21.0 | 4.7/7.0 | 4.7/6.9 |
| O$_3$ (ppbv) | 20 | 142 | 74 | 76 | 22/20 | 102/142 | 66/79 | 66/82 |
| CO (ppbv) | 167 | 1101 | 345 | 343 | 180/167 | 989/1101 | 348/344 | 335/347 |
| SO$_2$ (ppbv) | 0.1* | 7.1 | 0.6 | 0.4 | 0.1*/0.1* | 3.4/7.1 | 0.3/0.8 | 0.2/0.6 |
| PM$_{2.5}$ (µg m$^{-3}$) | 1* | 65 | 15 | 10 | 1*/1* | 65/61 | 12/17 | 8/13 |
| S$_a$ (m$^{-1}$) | 2.3E-6 | 1.2E-3 | 3.0E-4 | 2.5E-4 | 2.3/3.5E-6 | 12/8.3E-4 | 2.1/3.6E-4 | 1.7/3.4E-4 |
| S$_a$×f(RH) (m$^{-1}$) | 7.0E-6 | 3.5E-3 | 8.3E-4 | 7.1E-4 | 7.0/11E-6 | 3.5/2.5E-3 | 6.1/9.7E-4 | 4.9/9.0E-4 |
| p (kPa) | 83.5 | 84.7 | 84.4 | 84.5 | 84.3/83.5 | 84.7/84.7 | 84.5/84.4 | 84.5/84.5 |
| T (℃) | 14.8 | 24.4 | 19.7 | 19.7 | 14.8/17.4 | 22.0/24.4 | 18.6/20.4 | 18.9/20.3 |
| RH (%) | 46 | 100 | 96 | 100 | 77/46 | 100/100 | 98/95 | 100/100 |
| WD (º) | 5 | 356 | 170 | 186 | 10/5 | 356/349 | 160/179 | 167/195 |
| WS (m s$^{-1}$) | 0* | 18.5 | 5.1 | 4.4 | 0*/0.3 | 10.6/18.5 | 4.2/5.7 | 4.1/4.8 |
| J(NO$_2$) (10$^{-3}$ s$^{-1}$) | -/- | 7.9 | 1.2 | 2.2 | -/- | 7.6/7.9 | 1.0/1.3 | 0.18/0.24 |

*: near or below the detection limit of the used instrument

**Table 2: Summary of ground-based or aircraft-based HONO measurements at background/remote sites (including mountain or pole sites) and cities near Mt. Tai.**

| Location | Altitude (m) | Period | Technique | Mean (pptv) | Range (pptv) | HONO/NO$_x$ (%) | Reference |
|---|---|---|---|---|---|---|---|
| **Background sites** | | | | | | | |
| Zugspitze, Germany | 2650 | 9-16 June 2001 | LOPAP | 12.6[*] | 2-35 | 2.5[*] | (Kleffmann et al., 2002) |
| Cimone, Italy | 2165 | 8-17 May 2004 | Coil-HPLC | | 0-40 | | (Beine et al., 2005) |
| Hohenpeissenberg, Germany | 980 | 3-12 Jul 2002 29 Jun-14 Jul 2004 | Denuder-IC | 100[a]/30[b] | <10-200 | 6.3[c] | (Acker et al., 2006) |
| Whiteface, USA | 1483 | 14 Jun-20 Jul 1999 | Coil-HPLC | 46 | <5-400 | 23 | (Zhou et al., 2007) |
| Jungfraujoch, Switzerland | 3580 | 2-7 Nov 2005 | LOPAP | 7.5 | <0.5-50 | 4.6 | (Kleffmann and Wiesen, 2008) |
| Barrow, USA | ~3 | 13 Mar-14 Apr 2009 | LOPAP | 27 | <0.4-500 | 6.0 | (Villena et al., 2011) |
| Concordia station, Antarctic plateau (CSAP) | 3233 | 22 Dec 2010-18 Jan 2011 | LOPAP | 28 | 5-59 | | (Kerbrat et al., 2012) |
| CSAP | 3233 | 9-23 Feb 2011 | LOPAP | 3 | 0-14 | | (Kerbrat et al., 2012) |
| CSAP | 3233 | 4 Dec 2011-13 Jan 2012 | LOPAP | 35[d]/30[e] | | | (Legrand et al., 2014) |
| Southeastern US | PBL | 1 June-15 July 2013 | LPAP | 11.2 | 3-34 | | (Ye et al., 2018) |
| Southeastern US | FT | 1 June-15 July 2013 | LPAP | 5.6 | 1-15 | | (Ye et al., 2018) |
| Mt. Tai | 1534 | 1-31 Dec 2017 | LOPAP | 150 | 0-1140 | 3.2 | (Jiang et al., 2020) |
| Mt. Tai | 1534 | 5 Mar-8 Apr 2018 | LOPAP | 130 | 0.5-3230 | 6.0 | (Jiang et al., 2020) |
| Mt. Tai | 1534 | 9-31 Jul 2018 | LOPAP | 133 | 1-880 | 6.4 | This study |
| **Nearby cities** | | | | | | | |
| Jinan | ~150 | 26 Nov 2013-5 Jan 2016 | MARGA | 350 | <3340 | | (Wang et al., 2015) |
| Jinan | ~150 | 1 Sep 2015-31 Aug 2016 | LOPAP | 1150 | 17-8360 | 7.9[a]/5.6[b] | (Li et al., 2018) |
| Tai'an | ~150 | 29 May-8 Jul 2018 | LOPAP | 620 | 50-2970 | 4.2 | (Xue et al., 2022) |

[*]: data published in Kleffmann and Wiesen (2008) and some unpublished data from the study of Kleffmann et al. (2002).

[a]: noontime, [b]: night-time, [c]: HONO/NO$_2$, [d], and [e]: mean values in December and January, respectively.

PBL and FT: the planetary boundary layer and the free troposphere.

## 3.2 Impact of Anthropogenic Emissions on the Measured HONO

### 3.2.1 Impact of Emissions at the Summit Level (1534 m a.s.l.)

High values of NO$_x$/NO$_y$ were expected in a very fresh plume with significant local emissions. Throughout the campaign, the average NO$_x$/NO$_y$ ratio was $0.43 \pm 0.28$, which was much lower than fresh plumes observed in the nearest city of Tai'an with an average of $0.93 \pm 0.05$ (from the measurement at the foot station), indicating an aged air mass and a general small impact of nearby anthropogenic emissions at the summit level.

However, regular local emissions caused rapid increases of some pollutants. As an example, the most rapid increase of HONO and other pollutants, which was observed between 5:20 and 6:20 on 29[th] July 2018 is shown in Figure 4. During this event, HONO rapidly increased from 18 to 700 pptv, in concert with rises in NO, CO, PM$_{2.5}$, NO$_x$, and NO$_y$ but a decrease in O$_3$ (Table 3). The synchronous increase in NO (a primary pollutant of combustion) and the decrease in O$_3$ indicates a relatively fresh plume due to the fast titration reaction, as shown in R-1:

$$NO + O_3 \rightarrow NO_2 + O_2, \qquad k_1 \qquad \qquad \textbf{R-1}$$

During this event, air mass originated from the south (Figure 2), the polluted urban region (Figure S1E) rather than the direction of the potential sources at the summit level. This event lasted about 1.5 hours (5:20-6:50), much longer than the duration of the typical fresh plumes observed at the foot station. Furthermore, the $NO/NO_x$ ratio of this plume was 0.21, lower than that of the direct $NO/NO_x$ emission ratio of ~0.9 (Carslaw and Beevers, 2005; He et al., 2020; Kurtenbach et al., 2012; Wild et al., 2017). This is also lower than that of the close-to-fresh plumes observed at the foot station with an average $NO/NO_x$ ratio of $0.46\pm0.19$ at high $O_3$ levels (Xue et al., 2022). Therefore, we could conclude that the observed plume should originate from the foot urban region rather than nearby emissions at the summit. The $\Delta HONO/\Delta NO_x$ within this plume was 8%, much larger than that inferred from direct emissions (typically inferred as less than 1%). The ratio could be enhanced by: 1) night-time $NO_2$-to-HONO conversion at the ground level where the air mass was already aged before being transported to the summit level, 2) in-plume $NO_2$-to-HONO conversion along the mountain slope (rock and vegetation surfaces, etc.), and 3) in-plume $NO_2$-to-HONO conversion on particle surfaces as both the boundary layer height (BLH) elevation and the valley breeze are initialized after sunrise.

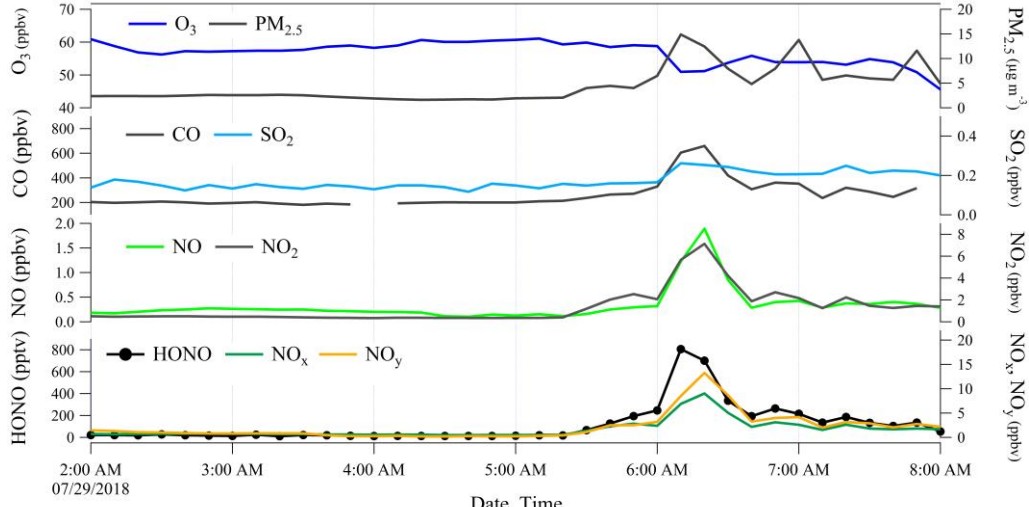

**Figure 4: HONO and related species measured on the morning of 29th July 2018.**

**Table 3: Concentrations of HONO and related species measured at 05:20 and 06:20 on 29th July 2018**

| Species | 05:20 | 06:20 | Δ |
|---|---|---|---|
| HONO/pptv | 18 | 700 | 682 |
| CO/ppbv | 214 | 659 | 445 |
| NO/ppbv | 0.1 | 1.9 | 1.8 |
| $NO_2$/ppbv | 0.4 | 7.1 | 6.7 |
| $NO_x$/ppbv | 0.5 | 9.0 | 8.5 |
| $NO_y$/ppbv | 0.4 | 13.3 | 12.9 |
| $PM_{2.5}$/µg m$^{-3}$ | 2.1 | 12.4 | 10.3 |
| $O_3$/ppbv | 59 | 51 | -8 |

### 3.2.2 Impacts from the Level Below the Summit

### 3.2.2.1 Insight on the Morning Peaks of the Diurnal Profiles

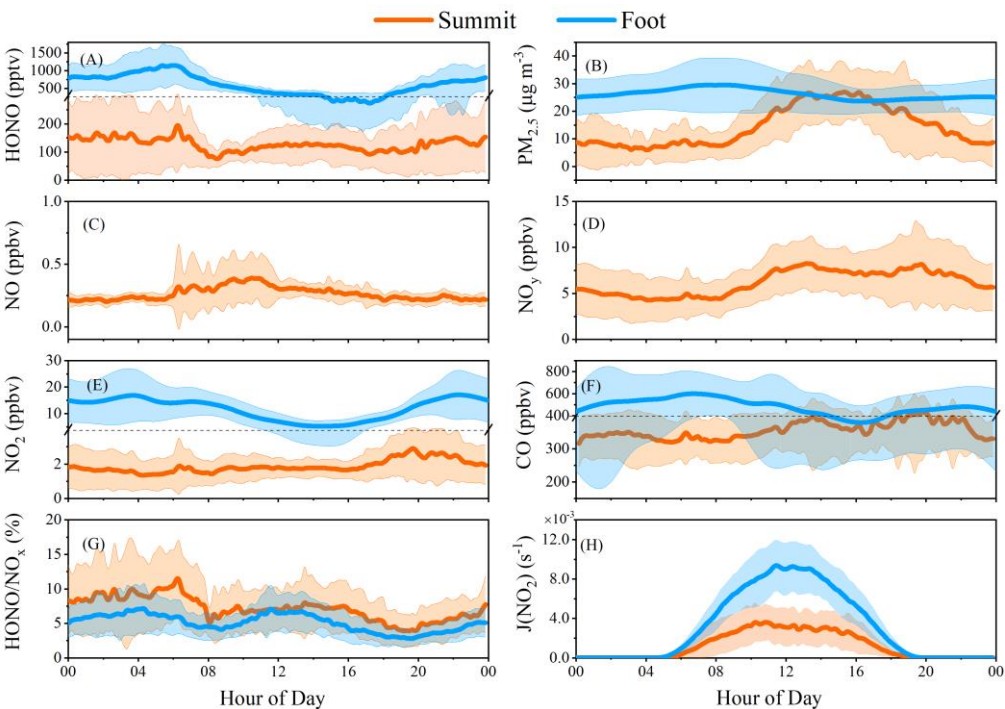

**Figure 5: Diurnal variations of HONO and related parameters observed at the summit (orange) and the foot (blue) stations. Data for the summit station have a time resolution of 10 min. Hourly $PM_{2.5}$, $NO_2$, and CO were available but NO and $NO_y$ were not available at the foot station during this period. All the data were in the same measurement period from 9th to 31st July, except for HONO, HONO/$NO_x$, and J($NO_2$) for the foot station measured from 29th May to 8th July.**

In Figure 5 the campaign averaged diurnal data is shown, in which most observed species, including HONO, NO, $NO_2$, $NO_y$, CO, and $PM_{2.5}$, showed small peaks during 6:00 – 6:30. This suggests a regular process responsible for this phenomenon rather than an accidental event. Note that the sun started to rise and heat the ground surface, as well as the mountain surface, one

hour before those peaks, leading to an increasing BLH (Anisimov et al., 2017). On the other hand, sunrise would initiate the
daytime upslope valley breeze wind (Kalthoff et al., 2000; Schmid et al., 2020; Ye et al., 1987), which could also be supported
by the increasing pressure and temperature (1 hour after sunrise) observed at the summit (Figure S2). Hence, it can be inferred
that the morning peaks resulted from the rising air parcel, within which pollutants accumulated during night-time. Interestingly,
similar morning peaks were also observed in winter and spring (Figure S3A), indicating the persistent impact of this process.

### 3.2.2.2 Insight on the Seasonal HONO Variations

In addition to the morning peaks analysis, seasonal HONO variations at the summit were also summarized (Figure 6 and S3),
including measurements in winter, spring, and summer. Distinctly higher $PM_{2.5}$ and $NO_2$ were observed in winter (Figure 6B
and 6C) than in summer. However, HONO levels in winter/spring/summer were similar (Figure 6A), indicating that the
aerosol-derived sources did not dominate HONO formation at the summit level. In general, HONO levels observed at the
ground level of the NCP were significantly higher in winter than in summer (Li et al., 2018; Nie et al., 2015; Xue et al., 2020).
A similar HONO level observed in summer was possible because of a more rapid vertical exchange between the ground level
and the summit level (see Section 3.2.3).

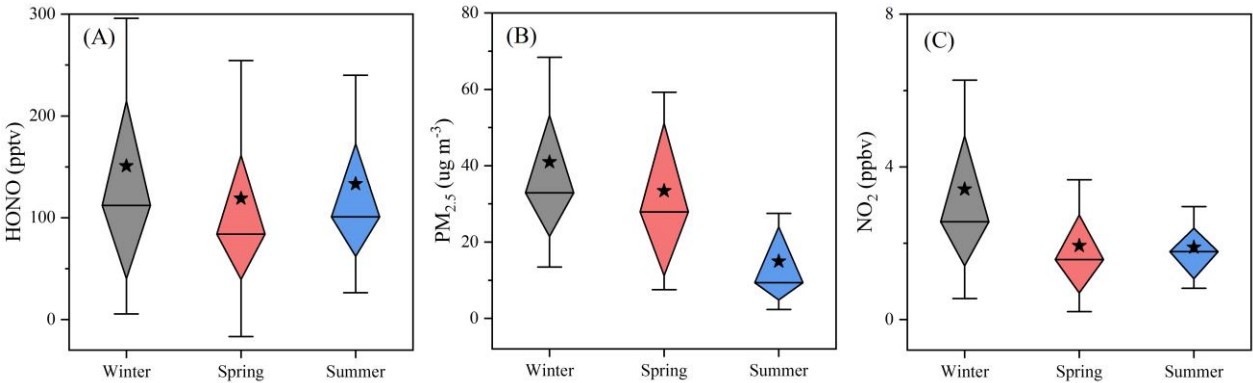

**Figure 6: Statistic summaries of (A): HONO, (B): $PM_{2.5}$, and (C): $NO_2$ in the three seasons. Error bars represent the standard
deviation. The top and the bottom of each diamond represent the 25% and 75% percentages, respectively. The star and the line
inside each diamond denote the average and the median, respectively. Data for winter and spring was taken from Jiang et al. (2020).**

### 3.2.2.3 Insight on the Comparison of Pollutants at the Foot and Summit Level

Comparison of daytime (5:00 – 18:00) average $PM_{2.5}$, CO, $O_3$, and $SO_2$ observed at the foot and the summit stations are shown
in Figure 7. It is apparent that all the average daytime levels of primary pollutants (CO and $SO_2$), partially primary pollutant
($PM_{2.5}$), and secondary pollutant ($O_3$) show very similar variation trends at both monitoring stations, revealing 1) a significant
or even dominant impact of pollutants at the foot level on that at the summit level, and 2) the presence of a pathway that
enables the vertical air mass exchange between the summit and the foot levels. This was also consistent with the higher daytime
HONO (Figure S3A) observed at the summit station in winter than in summer because the regional pollution was generally
much more severe in winter than in summer.

Besides, during night-time, the summit (~1500 m altitude) is above the boundary layer (in the residual layer), and similar variation trends of pollutants were also found at the foot and the summit stations (Figure S4), indicating still the presence of vertical air mass exchange at night. This could also be inferred from the higher night-time HONO (Figure S3A) in summer than in winter because 1) more south winds (the direction to Tai'an city) were observed in summer (Figure S5) and 2) the nocturnal boundary layer height was generally much lower in winter than that in summer.

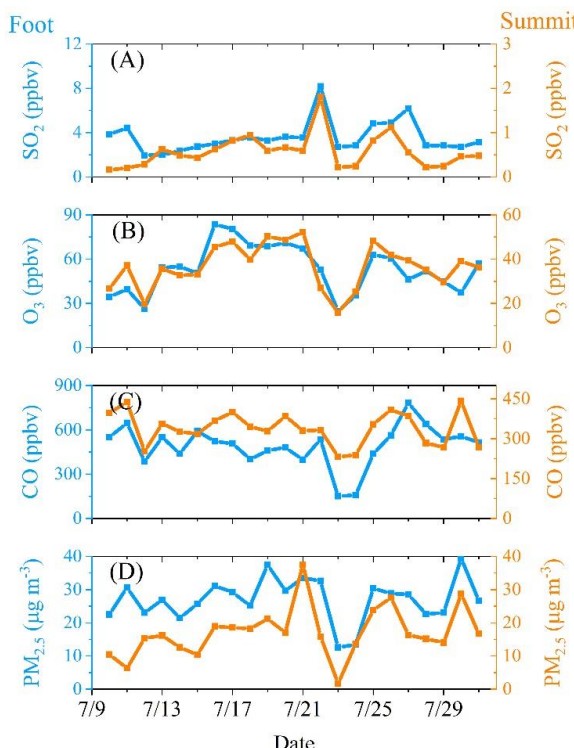

**Figure 7: Comparison of daytime (5:00 – 18:00) average (A): SO₂, (B): O₃, (C): CO, and (D): PM₂.₅ observed at the foot (left axis in blue) and summit (right axis in orange) stations during the same period from 9th to 31st July.**

### 3.2.3 Impact from Tai'an City (150 m a.s.l.)

Besides the discussion in Section 3.2.1, five arguments point to the potential impact from pollution in the nearest city (Tai'an city, ~150 m a.s.l.) on the summit HONO level:

a) the "∩" shape of HONO variation in the daytime was different from that of $NO_2$ (a constant level during the daytime), $NO_y$ (which increased in the early morning and then remained stable at noontime, followed by a continuous increase in the late afternoon) and $PM_{2.5}$ (which also showed a "∩" shape variation but its peak was 3 hours later than the HONO peak). These observations indicate that the observed HONO at the summit was not dominated by in-situ aerosol-derived formation (Figure 5) but an external HONO source such as transport;

b) high-level HONO was frequently observed at the ground level (150 m a.s.l.) in Tai'an city (Table 2), and almost the same variation trends of $HONO/NO_x$ were observed at both the summit and foot stations (Figure 5G);

c) HONO peaks at the summit occurred at noontime when the BLH was high, and valley breeze wind was strong;

d) high-level HONO (>200 pptv) observed at the summit mainly appeared when the air mass came from south or southwest (the direction to Tai'an city, see Figure 8);

e) HONO peaks occurred synchronously with the peaks of $SO_2$, which is mainly emitted at the ground level, and $NO_2$, which is an important HONO precursor (Figure 3).

The impact could be achieved through: 1) the air mass ascending by valley breeze upslope wind (daytime) and by the north wind (daytime and night-time, Section 3.2.2.3), and 2) HONO formation during the air mass ascending process, i.e., HONO formation through the $NO_2$ heterogeneous uptake on the mountain slope surfaces (George et al., 2005; Marion et al., 2021;

Stemmler et al., 2006). The HONO production from the above processes was defined as P(HONO)$_{transport}$ and will be discussed in Section 3.6.

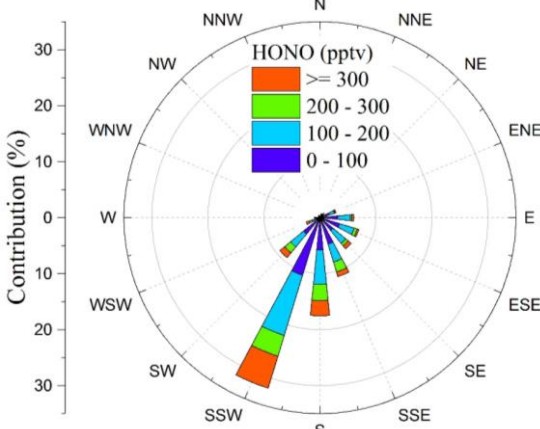

**Figure 8: Pollution rose plot of HONO against the wind direction. The frequency contribution of counts by wind direction is also shown on the left axis.**

The length of the hypotenuse from the foot to the summit is about 4.2 km, with an average elevation angle of about 20°. In the daytime, the valley breeze could occur with an upslope wind speed of 2 – 5 m s$^{-1}$ reported in previous measurements (Kalthoff et al., 2000; Schmid et al., 2020; Ye et al., 1987; also see https://glossary.ametsoc.org/wiki/Upvalley_wind), it takes about 14 – 35 min (t$_{transport}$) for the air mass to be transported from the foot to the summit. The upslope valley breeze wind could transport polluted air mass from the foot to the summit levels. This process could be accelerated by the dominant south wind (Figure 8)

as the urban site (150 m a.s.l.) is south of the summit station (1534 m a.s.l.). The mean south winds measured at the ground and summit stations are >2 and >5 m s$^{-1}$, respectively. Then the integrated wind speed along the mountain slope should be 4 – 10 m s$^{-1}$, and the calculated t$_{transport}$ will be reduced to 7 – 17.5 min.

The key question is the quantity of HONO that still exists after transport from the foot to the summit levels regarding its photolysis in the daytime. Assuming first-order decay of HONO by photolysis during the transport, the remaining HONO and

ratio at the summit can be calculated:

$$c_t = c_0 e^{-J(HONO)*t_{transport}}, \hspace{3cm} \textbf{Eq-1}$$

$$\alpha = \frac{c_t}{c_0} = e^{-J(HONO)*t_{transport}},\qquad\qquad\qquad\qquad\qquad\qquad\qquad\qquad\textbf{Eq-2}$$

where $c_t$, $c_0$, J(HONO), and $\alpha$ represent the remaining HONO after a transport period ($t_{transport}$), the initial HONO concentration at the foot, the HONO photolysis frequency, and the remaining proportion of HONO.

Figure 9 shows the calculated $\alpha$ with $t_{transport}$ = 7 or 17.5 min during the daytime. It is apparent that $\alpha$ is larger than 43% with $t_{transport}$ = 17.5 min and larger than 72% with $t_{transport}$ = 7 min, providing a theoretical basis for the potential role of vertical HONO transport from the ground to the summit levels. The calculations don't consider the atmospheric dilution or dispersion during the transport, which may reduce $\alpha$. This effect could be roughly quantified by comparing levels of long lifetime species (e.g., CO) at the foot and the summit stations. Hourly CO averages at noon are 493 and 379 ppbv measured at the foot and the

summit stations, respectively (Figure 5). This preliminarily indicates a dilution factor of 1.3. The dilution process may also similarly affect HONO, i.e., $\alpha$ is expected to be reduced by a factor of 1.3, leading to $\alpha$ values of >55% and >33% with $t_{transport}$ = 7 or 17.5 min, respectively. The above calculation only included the daytime HONO sink through photolysis and atmospheric dilution, but the sources, such as NO + OH and heterogeneous $NO_2$ reactions, were not considered, and hence, the calculated $\alpha$ represents a lower limit. Thus, the impact of transport was expected to be larger when 1) taking other HONO formation paths

(e.g., $NO_2$ heterogeneous reactions on the mountain surfaces and the vegetation surfaces) into account, and 2) vegetation shadows on the mountain surface slow down HONO photolysis during the transport. Therefore, ground level (~150 m a.s.l.) HONO as well as its formation during transport may affect the HONO measurement at the summit significantly. The quantification of the contribution will be discussed in Section 3.6.

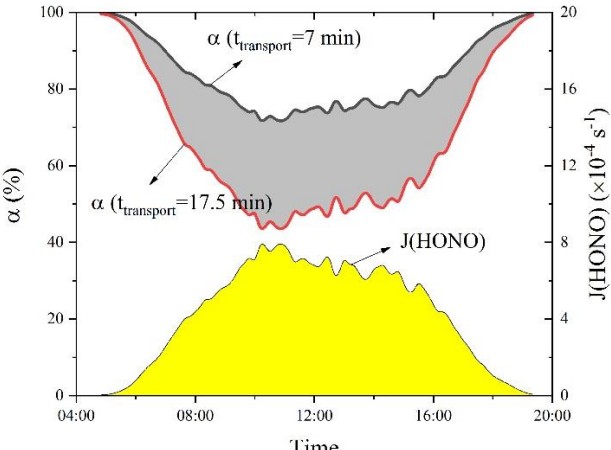

**Figure 9: Diurnal profiles of the remaining proportion of HONO ($\alpha$) after a period of transport ($t_{transport}$) from the ground to the summit levels and J(HONO) during the daytime.**

### 3.3 Daytime Unknown HONO Source Strength

The photo-stationary state (PSS), presented by the following equations, is valid to calculate the unknown HONO source strength ($P_{un}$) when local emission was negligible (Crilley et al., 2016; Kleffmann et al., 2005; Michoud et al., 2012). The

predicted HONO concentration by PSS ([HONO]$_{PSS}$) and $P_{un}$ could be calculated by Eq-3 and Eq-4, respectively.

$$HONO + h\nu \rightarrow NO + OH, \qquad\qquad J(HONO), \qquad\qquad\qquad\qquad \text{R-2}$$

$$NO + OH \rightarrow HONO, \qquad\qquad k_1 = 9.8 \times 10^{-12} \ cm^3 \ molecule \ s^{-1} \qquad \text{R-3}$$

$$HONO + OH \rightarrow NO_2 + H_2O, \qquad k_2 = 6.0 \times 10^{-12} \ cm^3 \ molecule \ s^{-1} \qquad \text{R-4}$$

$$[HONO]_{PSS} = \frac{k_1 \times [NO] \times [OH]}{J(HONO) + k_2 \times [OH]}, \qquad\qquad\qquad\qquad \text{Eq-3}$$

$\quad P_{un} = ([HONO] - [HONO]_{pss}) \times (J(HONO) + k_2 \times [OH]), \qquad\qquad \text{Eq-4}$

OH measurements were not available during this campaign. One popular used method to estimate OH is based on the correlation between OH and solar ultraviolet radiation (e.g., $J(O^1D)$) (Rohrer and Berresheim, 2006). Considering that Mt. Tai is surrounded by polluted regions, empirical formulas between OH and $J(O^1D)$ from the ground or other mountain measurements, may not be reasonable here. In June 2006 Kanaya et al. conducted a comprehensive field campaign at the

summit of Mt.Tai (Kanaya et al., 2009). OH levels and sources were studied by a box model. From the average diurnal variations of the modeled OH and $J(O^1D)$, a significant correlation ($R^2>0.9$) between them was found, which was used here to estimate OH concentrations. This method could lead to some uncertainties in OH levels due to: 1) high correlations between OH and $J(O^1D)$ was found in their average diurnal variations but may not be in their time series, and 2) HONO was not constrained in the box model simulations so that OH could be underestimated for the MTX 2006 campaign.

One way to consider HONO impact is to discuss the OH uncertainties caused by the lack of HONO chemistry. Assuming that the impact of HONO on OH levels is determined by its contribution to primary OH production, we can preliminarily deduce the OH uncertainties at the summit station caused by the lack of HONO chemistry based on measurements and model simulations for the foot station. At the foot station, HONO photolysis made a contribution of 64% to primary OH production (see Section 3.6). If the box model was not constrained by the measured HONO, OH would be underestimated by 25% (see

the companion paper, Xue et al. (2022)). At the summit station, HONO contributed 18% of the primary OH production (see Section 3.6). Therefore, OH underestimation due to the lack of HONO chemistry at the summit station should be roughly around 7% or so.

Hence, to cover the uncertainties caused by the above issues, we added OH sensitivity tests, reducing or increasing the OH level by 30%, to quantify the impact of the OH uncertainties in our further analysis and conclusion. The used OH, the

corresponding $HONO_{pss}$, $P_{un}$ and results from the sensitivity tests were also shown in Figure S6. The estimated OH level was lower than that modeled during the MTX campaign (Kanaya et al., 2009, 2013). This is mainly caused by lower $J(O^1D)$ resulting from frequent cloudy weather during the present study period. For instance, the average RH during this campaign was 96%, which is much higher than that during the MTX campaign (67%). Reducing or enlarging OH levels by 30% indeed remarkably impact $HONO_{pss}$. However, $HONO_{pss}$ (5-15 pptv level) is still 1-2 orders of magnitude lower than the observed

HONO (50-200 pptv level), leading to a small impact of variable OH and $HONO_{pss}$ levels on $P_{un}$. Hence, we highlight the uncertainties in OH levels estimated by the current method, but its impact on following HONO budget analysis should be small as discussed above.

The diurnal variation of the calculated noontime (10:00 – 16:00) $P_{un}$ is shown in Figure 10. Campaign-averaged $P_{un}$ was about 290 ± 280 pptv h$^{-1}$ with a maximum of about 1800 pptv h$^{-1}$. The maximum $P_{un}$ value appeared at midday (13:00), indicating a photo-enhanced HONO source. Similarly, high correlations (r = 0.79, 0.83, or 0.83) were found between $P_{un}$ and $NO_2 \times J(NO_2)$, $pNO_3 \times J(HNO_3)$, or $NO_y \times J(HNO_3)$ (Table 4), suggesting the potential HONO formation from photosensitized $NO_2$ reactions or photolysis of $NO_z$ ($NO_z = NO_y – NO – NO_2$) species such as particulate nitrate ($pNO_3$). Moreover, the relatively poor correlations (r = 0.17 or 0.64) between $P_{un}$ and $NO_2 \times S_a$ or $NO_2 \times S_a \times J(NO_2)$ (Table 4) suggested minor roles of dark and photo-enhanced $NO_2$ uptake on the aerosol surface in the HONO formation. Besides, a high correlation between $P_{un}$ and HONO (r = 0.76) was obtained. A possible reason could be that HONO and other pollutants were not dominated by in situ formation but by transport, as discussed in Section 3.2.2. As correlation analysis is only a preliminary indicator and it might not be instructive for HONO budget analysis when the vertical air mass exchange occurs, further investigation of $NO_2$ uptake on aerosol surface and photolysis of $pNO_3$ is presented in Sections 3.4 and 3.5, respectively.

**Table 4: The correlation coefficients (r) between HONO or $P_{un}$ and other parameters (first column).**

| Correlations | HONO | $P_{un}$ |
|---|---|---|
| CO | 0.20 | -0.10 |
| NO | 0.06 | 0.09 |
| $NO_2$ | 0.53 | 0.05 |
| $NO_x$ | 0.52 | 0.06 |
| $NO_y$ | 0.49 | 0.45 |
| $NO_z$ | 0.38 | 0.49 |
| $PM_{2.5}$ | 0.07 | 0.21 |
| HONO | -- | 0.76 |
| HONO*J(HONO) | -- | 0.988 |
| $J(NO_2)$ | -0.04 | **0.77** |
| $NO_2 \times PM_{2.5}$ | 0.28 | 0.25 |
| $NO_2 \times J(NO_2)$ | 0.03 | **0.79** |
| $pNO_3 \times J(HNO_3)$ | 0.05 | **0.83** |
| $NO_y \times J(HNO_3)$ | 0.06 | **0.83** |
| $NO_2 \times S_a$ | 0.47 | 0.17 |
| $NO_2 \times S_a \times J(NO_2)$ | 0.10 | 0.64 |

### 3.4 Constraint on HONO Formation from NO₂ Uptake on the Aerosol Surface

During the daytime, the HONO production rate from the $NO_2$ uptake on the aerosol surface ($P(HONO)_a$) with the photo-enhanced effect was parameterized by the following equation. Note that dark $NO_2$ uptake on the aerosol surface was not considered due to a much lower uptake coefficient generally at a level of $10^{-6}$ (George et al., 2005; Han et al., 2017; Stemmler

et al., 2006, 2007).

$$P(HONO)_a = \frac{v(NO_2) \times S_a \times [NO_2]}{4} \times [\gamma_a \times \frac{J(NO_2)_{measured}}{0.005 \ s^{-1}}], \quad\quad\quad \text{Eq-5}$$

where $v(NO_2)$, $S_a$, $\gamma_a$, $J(NO_2)_{measured}/0.005$ are the molecular speed of $NO_2$ (m s$^{-1}$), aerosol surface density (m$^{-1}$), the photo-enhanced uptake coefficient of $NO_2$ on the aerosol surfaces, and the photo-enhancement factor normalized to a $J(NO_2) = 0.005$ s$^{-1}$. In Eq-5, an upper limit HONO yield for the $NO_2$ conversion of 100% was assumed. Additionally, RH was proposed to

significantly influence aerosol surface density, especially at our site, with frequently high RH up to 100%. Then besides calculating the aerosol surface density based on the measured aerosol size distribution ($S_{a\_measured}$), we estimated the effective aerosol surface density in m$^{-1}$ with an RH enhancement factor $f(RH)$ ($S_a = S_{a\_measured} \times f(RH)$) using the following equation:

$$f(RH) = 1 + a \times (RH/100)^b, \quad\quad\quad \text{Eq-6}$$

where a and b are empirical values of 2.06 and 3.60, respectively (Liu et al., 2008). The average $S_a$ without and with RH

enhancement is $3.0 \times 10^{-4}$ and $8.3 \times 10^{-4}$ m$^{-1}$, respectively. Note that the uncertainty of $S_a$ is not expected to cause a significant

uncertainty on HONO budget analysis as $P(HONO)_a$ was not the dominant source (Figure 10 and see the below discussion on $P(HONO)_a$ contribution).

As $P_{un}$ includes all the sources except NO + OH, then $P(HONO)_a \ll P_{un}$ can always be obtained. Hence, the real $\gamma_a$ value should be much lower than the inferred ones ($\gamma_{a\_inferred}$) from $P_{un} = P(HONO)_a$. In total, 606 $\gamma_a$ values were inferred based on the measurements, varying from $1.3\times10^{-4}$ to $8.5\times10^{-3}$, with a mean of $(8.3 \pm 7.5)\times10^{-4}$. However, the minimum ($\gamma_{a\_inferred\_mini}$) of $1.3\times10^{-4}$ is still very high, compared to the results of most lab studies, in which values of $\gamma_a$ of typically at few times $10^{-5}$ or even less were observed (Han et al., 2016; Liu et al., 2019a; Ndour et al., 2008; Sosedova et al., 2011; Stemmler et al., 2007). Hence a popularly used value of $\gamma_a = 2\times10^{-5}$ was used to calculate $P(HONO)_a$ and $\gamma_{a\_inferred\_mini}$ of $1.3\times10^{-4}$ was also used for uncertainty analysis as the upper limit.

The calculated $P(HONO)_a$ with these $\gamma_a$ values are shown in Figure 10. It is obvious that $P(HONO)_a$ is significantly lower than $P_{un}$ with either lab-based $\gamma_a = 2\times10^{-5}$ or even $\gamma_{a\_inferred\_mini} = 1.3\times10^{-4}$, pointing out the minor role of $NO_2$ uptake on the aerosol surface in daytime HONO formation. With the lab-based $\gamma_a = 2\times10^{-5}$, $P(HONO)_a$ could only explain 3% of $P_{un}$, which is similar to previous model studies (Liu et al., 2019b; Xue et al., 2020; Zhang et al., 2016). The contribution of $P(HONO)_a$ to $P_{un}$ increased when using $\gamma_{a\_inferred\_mini}$, but resulted from an overestimated $\gamma_a$ as discussed before. Nevertheless, analysis in this study still could be an important effort in the field constraints on the $NO_2$-to-HONO conversion on the aerosol surface.

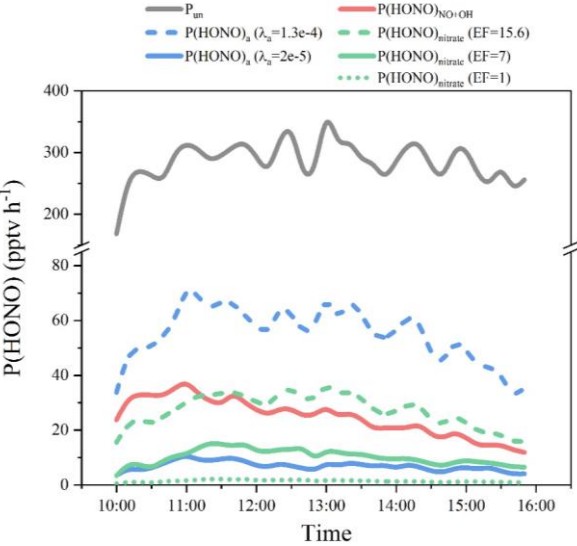

**Figure 10: Unknown HONO source strength ($P_{un}$), and HONO production rates from NO + OH ($P(HONO)_{NO+OH}$), $NO_2$ uptake on the aerosol surface ($P(HONO)_a$) with the inferred $\gamma_{a\_inferred\_mini} = 1.3\times10^{-4}$ and the popular used $\gamma_a = 2\times10^{-5}$), and nitrate photolysis ($P(HONO)_{nitrate}$) with EF values of 1, 7 and 15.6.**

### 3.5 Constraint on HONO Formation from the Photolysis of Particulate Nitrate

As one of the important inorganic components of aerosols, particulate nitrate ($pNO_3$) could undergo photolysis, with the production of HONO. This process needs more field constraints as discussed in the Introduction section. During the present

campaign, pNO₃ at the summit was measured by a filter method every 2 hours (Liu et al., 2020), but it suffered a sampling problem after 12th July. Because $NO_z$ ($NO_z = NO_y-NO-NO_2$) mainly contains $pNO_3$ and its precursors, e.g., $HNO_3$ and $N_2O_5$,

similar variations were expected between $NO_z$ and $pNO_3$. As shown in Figure S7, $NO_z$ and $pNO_3$ exhibited a very high correlation ($R^2 = 0.895$), for which $pNO_3$ makes 44% of $NO_z$ and this fraction was used to estimate $pNO_3$ in the period when it was not measured. The uncertainty of the estimated $pNO_3$ should have no significant impact on daytime HONO formation concerning its small contribution to daytime HONO formation (see Section 3.6).

A high correlation between $P_{un}$ and $pNO_3 \times J(HNO_3)$ was found (Table 4), suggesting a possible impact of $pNO_3$ on HONO

formation. But one should bear in mind that the high correlation might also be caused by the remarkable impact on $pNO_3$ formation from HONO-related reactions (e.g., $HONO \xrightarrow{h\nu} OH \xrightarrow{NO_2} HNO_3 \rightarrow pNO_3$) (Xue et al., 2020) or other photolytic processes. For parameterization, an enhancement factor (EF) was defined as the ratio of photolysis frequencies of $pNO_3$ to gas-phase $HNO_3$. Then HONO production from $pNO_3$ photolysis ($P(HONO)_{nitrate}$) could be quantified by Eq-7:

$$P(HONO)_{nitrate} = pNO_3 \times J(HNO_3) \times EF, \qquad\qquad \textbf{Eq-7}$$

Similar to $NO_2$ uptake on the aerosol surface, one can always find $P(HONO)_{nitrate} << P_{un}$. Hence, it is expected that the real EF should be much lower than the inferred ones ($EF_{inferred}$) from $P(HONO)_{nitrate} = P_{un}$. Therefore, 606 EF values were inferred, in the range of 15.6 to 1072, with a mean of $173 \pm 98$, which is much higher than those (around 1) determined in recent flow tube or smog chamber studies (Shi et al., 2021; Wang et al., 2021). The minimum ($EF_{inferred\_mini} = 15.6$) is at a similar level to field studies of Romer et al. (2018) and Zhou et al. (2003), and the lower values in the laboratory studies (Bao et al., 2018; Ye et

al., 2016, 2017; Zhou et al., 2011). To quantify the HONO production from $pNO_3$ photolysis, the EF value of 7 from a recent field study (Romer et al. 2018) was used for $P(HONO)_{nitrate}$ calculation, and $EF_{inferred\_mini}$ (15.6) from this study and EF values of ~1 from recent laboratory studies (Shi et al., 2021; Wang et al., 2021) were also used for the uncertainty analysis and comparison.

The calculated $P(HONO)_{nitrate}$ with these EF values is shown in Figure 10. With the EF = 7, $P(HONO)_{nitrate}$ was at a level of

half of NO + OH but much lower than $P_{un}$, which was also observed at the summit of Mt. Whiteface (Zhou et al., 2007). $P(HONO)_{nitrate}$ could explain 4.3% of the observed $P_{un}$. Its contribution varied from 0.6% to 9.6%, depending on EF values varying from 1 to 15.6 in the sensitivity tests. Therefore, with a $P(HONO)_a$ ($\gamma_a = 2 \times 10^{-5}$) contribution of 3% (Section 3.4) and a $P(HONO)_{nitrate}$ (EF = 7) contribution of 4.3%, the other sources (defined as $P(HONO)_{other} = P_{un} - P(HONO)_a - P(HONO)_{nitrate}$, mainly transported from the ground level as discussed in Section 3.2.2), made a dominated contribution of 92.7% to the

observed $P_{un}$.

Moreover, significant differences between EF values obtained from field studies and laboratory studies indicate a complex process of $pNO_3$ photolysis that may be influenced by various environmental parameters, e.g., the aerosol $pNO_3$ loading and the aerosol composition (Bao et al., 2018, 2020; Ye et al., 2016, 2017), and experimental laboratory conditions, e.g., collected particles on the filter or generated airborne particles (Shi et al., 2021; Wang et al., 2021). We, therefore, suggest that this

process still needs further field or laboratory constraints. To our knowledge, this study provided the first field constraint on the

aerosol-derived HONO sources in the NCP region, where the abundance of aerosol was frequently observed, and its role in HONO formation is still highly controversial.

The landscape (e.g., mountains) enhances the vertical air mass exchange, leading to a weak vertical HONO distribution within the boundary layer, which is not yet considered in previous studies (Jiang et al., 2020). This will underestimate the role of
ground-derived sources in HONO formation in the upper boundary layer over mountain regions. Radicals, including OH and $HO_2$, are not expected to be transported far due to their short enough lifetimes (<100 s). However, 15% of daytime HONO was formed at the ground level through NO + OH as reported in the companion ACP paper (Xue et al., 2022), and part of OH consumed at the ground level would be released at the summit level through HONO photolysis. Hence, it could be preliminarily inferred that radicals (i.e., OH) could be transported through their precursors/reservoirs (like HONO) with lifetimes longer
than themselves. Furthermore, the enhanced vertical air mass exchange could also lead to fast transport of other pollutants ($PM_{2.5}$, $O_3$, CO, $SO_2$, etc.) from the ground to the summit levels, which will significantly impact the atmospheric composition as well as its chemistry in the upper boundary layer or the residual layer. The discussion and implications in this study are instructive for further field and model studies.

### 3.6 Role of HONO in the Oxidizing Capacity of the Lower and the Upper Boundary Layer

$O_3$ was typically the major OH source at high altitude regions, including the upper boundary layer. Then we compared the OH production rates from $O_3$ and HONO photolysis to investigate whether HONO could play a significant role in the oxidizing capacity of the atmosphere at this high-altitude site. Photolysis of HONO and $O_3$ with their net OH production is shown in R-2 and R-5 to R-7, respectively. OH loss through HONO + OH and NO + OH was subtracted from $P(HO_x)_{HONO}$ to obtain $P(HO_x)_{HONO\_net}$.

$$O_3 + hv \rightarrow O_2 + O(^1D), \qquad J(O(^1D)) \qquad\qquad \text{R-5}$$

$$O(^1D) + H_2O \rightarrow 2OH, \qquad k_3 \qquad\qquad \text{R-6}$$

$$O(^1D) + M\ (N_2\ or\ O_2) \rightarrow O(^3P) + M\ (N_2\ or\ O_2), \qquad k_4 \qquad\qquad \text{R-7}$$

$$P(HO_x)_{HONO\_net} = [HONO] * J(HONO) - k_1 * [NO] * [OH] - k_2 * [HONO] * [OH], \qquad\qquad \text{Eq-8}$$

$$P(HO_x)_{O_3} = [O_3] * J(O(^1D)) * \phi, \qquad\qquad \text{Eq-9}$$

where the reaction constants were taken from the IUPAC kinetic database (https://iupac-aeris.ipsl.fr). The atmospheric RH and temperature largely influenced the branching ratio of R-6 to R-7. The average OH yield ($\phi$) during the campaign of 20% was used for calculating OH production from $O_3$ photolysis.

Additionally, in the companion paper in which HONO was reported to be the most important primary OH source at the foot station (Xue et al., 2022). A comparison between the role of HONO at the foot and the summit stations could provide more
insights into the importance of HONO throughout the boundary layer. Moreover, as reported in the companion paper, HONO observed at the foot station was mainly produced through $NO_2$ heterogeneous reactions and NO+OH. Therefore, the comparison could also shed light on the link between the atmospheric oxidizing capacity in the lower and the upper boundary

layer, although measurements at two stations were conducted during two consecutive periods rather than the same one in summer 2018.

Figure 11 displays the diurnal profiles of net OH production rates from HONO and $O_3$ photolysis at the foot and the summit stations. It is apparent that both $P(OH)_{HONO\_net}$ and $P(OH)_{O_3}$ showed higher levels at the foot station compared to the summit station. For instance, average $P(OH)_{HONO\_net}$ and $P(OH)_{O_3}$ at the foot station are 0.9 and 0.5 ppbv $h^{-1}$, respectively, both of which are significantly higher than those (0.06 and 0.28 ppbv $h^{-1}$) at the summit station. This is caused by relatively lower HONO and $O_3$ concentrations and lower solar photolysis frequencies as a result of frequent cloud formation observed at the

summit station.

In particular, after night-time accumulation, HONO photolysis is found to initialize daytime photochemistry in the early morning at the ground level (Alicke et al., 2002; Kleffmann, 2007; Platt et al., 1980). This was also observed at the foot station. As shown in Figure S8, at the foot station, the contribution of $P(OH)_{HONO\_net}$ to $P(OH)_{sum}$ was almost 100% at sunrise around 5:00. It showed a declining trend but still played the dominant role in $P(OH)_{sum}$, with a contribution larger than 90% in the

early morning (5:00-7:00). At the summit station, at 5:00, solar radiation was very weak, for instance, $J(NO_2)$ was only $3.6 \times 10^{-4}$ $s^{-1}$. At this time, $P(OH)_{HONO\_net}$ was slightly negative ($-7 \times 10^{-3}$ ppbv $h^{-1}$) due to OH loss through HONO + OH and NO + OH. $O_3$ photolysis was initialized at the same time, but $P(OH)_{O_3}$ was nearly zero ($7 \times 10^{-4}$ ppbv $h^{-1}$). From 6:00 to 7:00, a considerable amount of net OH was produced through HONO photolysis (0.04-0.09 ppbv $h^{-1}$), with its contribution to $P(OH)_{sum}$ decreasing from 64% to 39% (Figure S8). Hence, it could be inferred that daytime atmospheric photochemistry at the summit level is also

initialized by HONO photolysis.

On average, the contribution of $P(OH)_{HONO\_net}$ to $P(OH)_{sum}$ was 64% at the foot station, higher than that at the summit station (18%), indicating the essential role of HONO in the atmospheric oxidizing capacity at both the ground (lower boundary layer) and the summit (upper boundary layer) levels in mountainous regions. As discussed before, the transport from the ground to the summit levels contributed to the majority of HONO observed at the summit level. This points to a new insight that ground-

derived HONO played an important role in the oxidizing capacity, not only at the ground level but also in the upper boundary layer (~1500 m) in mountainous regions. Yet this vertical exchange might be only valid in the mountainous areas, and the follow-up regional impact still needs to be quantified by further model studies.

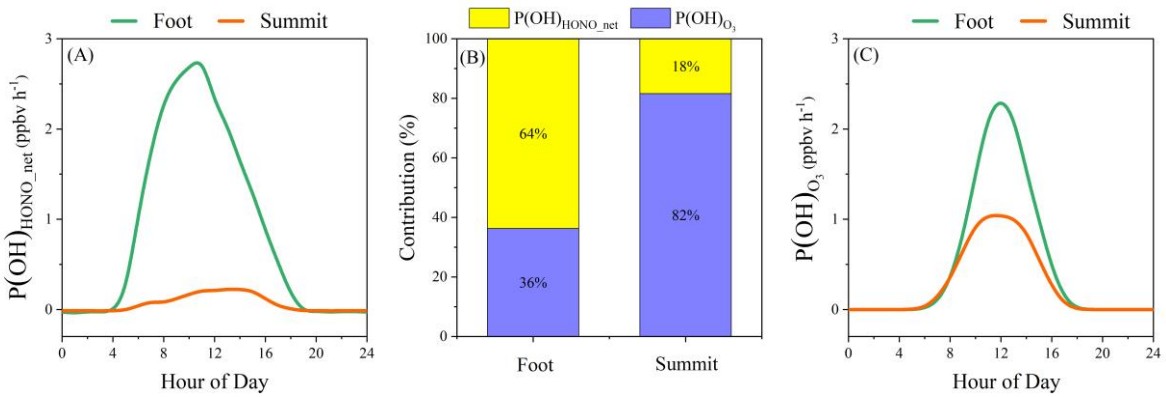

**Figure 11:** OH production from photolysis of HONO (P(OH)$_{HONO\_net}$) and O$_3$ (P(OH)$_{O_3}$) at the foot and the summit of Mt. Tai. (A): P(OH)$_{HONO\_net}$, (B): relative contributions, and (C): P(OH)$_{O_3}$.

## 4 Summary

Observations of HONO and related parameters at the summit of Mt. Tai (1534 m a.s.l.) in July 2018 were presented. The average HONO mixing ratio is $133 \pm 106$ pptv, with a maximum of 880 pptv, significantly higher than observations at other mountain summits worldwide. Along with observations at the ground level (the nearest city, Tai'an city), HONO formation from different paths and its role in the atmospheric oxidizing capacity of the upper boundary layer were explored and discussed. The main conclusions are listed as follows:

1. Constraints on the kinetics of NO$_2$ uptake coefficient on the aerosol surface and photolysis of pNO$_3$ were obtained based on the assumption that P$_{un}$ could be solely explained by NO$_2$ uptake on the aerosol surface, P(HONO)$_a$ or particulate nitrate photolysis, P(HONO)$_{nitrate}$. The inferred $\gamma_a$ and EF values were much higher than most values obtained from recent laboratory studies, indicating that the aerosol-derived HONO could not explain the observed P$_{un}$. In the NCP region, the abundance of aerosol was frequently observed, but its role in HONO formation is still highly controversial as a result of uncertain kinetics. This study provided the first field constraints on aerosol-derived HONO sources in this region and will be instructive for further laboratory or model studies.

2. With a $\gamma_a$ value of $2\times10^{-5}$ and an EF value of 7, P(HONO)$_a$ and P(HONO)$_{nitrate}$ showed small contributions (3% and 4.3%, respectively) to daytime HONO formation at the summit station. Both P(HONO)$_a$ and P(HONO)$_{nitrate}$ varied from negligible to moderate levels (similar to NO + OH), depending on $\gamma_a$ and EF values, suggesting the necessity to further study the related kinetics. Additionally, although high values of $\gamma_a$ ($1.3\times10^{-4}$) and EF (15.6) compared with recent studies were tested here, both sources were still much lower than the observed P$_{un}$. The remaining majority (92.7%) of P$_{un}$ was dominated by the rapid vertical transport from the ground to the summit levels including heterogeneous HONO formation on surfaces of the mountain slope, which was inferred from comprehensive evidence presented in this study.

3. A comparison of HONO contributions to primary OH at the summit and the foot levels was conducted. It was confirmed that HONO photolysis initialized daytime photo-chemistry at both sites in the early morning. On average, HONO made contributions of 64% and 18% to $P(OH)_{sum}$ at the foot and the summit levels, respectively, indicating the important role of HONO in the oxidizing capacity of the atmosphere in mountainous areas. HONO formation at the ground level could significantly influence the HONO mixing ratios and the atmospheric oxidizing capacity at the summit level through the vertical air mass exchange. Moreover, the enhanced vertical air mass exchange could also lead to a fast exchange of other pollutants between the ground and the summit levels, which significantly impacts the atmospheric composition as well as the chemistry in the upper boundary layer or the residential layer. However, those follow-up impacts, by far, are not quantified by the current model studies.

## Acknowledgment

We are grateful to Xiaowei He, Pengfei Liu, Chao Zhu, Jiarong Li, Hui Chen, Xianmang Xu, Hongyong Li, Pengcheng Zhang, and Jinhe Wang for their help on the measurement at the summit of Mt. Tai. We thank all researchers involved in this campaign from the Research Centre for Eco-Environmental Sciences-Chinese Academy of Sciences, Fudan University, Shandong Jianzhu University, Shandong University, and the Municipal Environmental Protection Bureau of Tai'an. We thank Yunqiao Zhou for his help with the map plotting. We thank the two anonymous reviewers and the editor, Yugo Kanaya, for their efforts towards improving our manuscript.

**Funding**: This work was supported by the National Natural Science Foundation of China (Nos. 91544211, 41727805, 41931287, 21976190, and 41975164), and the PIVOTS project provided by the Region Centre − Val de Loire (ARD 2020 program and CPER 2015 − 2020).

**Author Contribution**: C.X., C.Y., C.Z., Y.Z., H.L., and Z.G. performed the field measurements. C.X. analyzed the observation data and wrote the paper with inputs from all co-authors. C.Y. and J.K. helped with the data analysis and manuscript writing. J.K., C.Y., V.C., A.M., L.X., J.C., K.L., F.B., and Y.M. revised the manuscript.

**Competing Interests**: The authors declare no competing financial interest.

**Data Availability**: All the summertime data used in this study is available upon request from the corresponding authors.

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
