# Peer review of "Atmospheric Measurements at Mt. Tai - Part I: HONO Formation and Its Role in the Oxidizing Capacity of the Upper Boundary Layer"

_Atmospheric Chemistry and Physics, 2021_

## Author Comment (AC1)

**Response to RC1**

The manuscript presents results of a comprehensive field campaign at two different altitudes, the foot (150 m a.s.l.) and the summit (1534 m a.s.l.) of Mt. Tai (Shandong province, China). Mt. Tai locates in the middle of the NCP with a relatively high pollution level. The measured HONO diurnal profile shows a daytime peak at 12:30 local time, which is interesting since HONO diurnal profiles would typically peak during the night and early morning in more polluted regions. The topic is of interest to the scientific community and is suitable for publication in ACP after addressing the comments below.

The authors claim that OH+NO gas-phase reaction accounts for only 8% of measured HONO, and that 70-98% of the unknown HONO sources can be attributed to vertical transport from ground surfaces. However, the authors didn't show/present the OH values used to calculate the OH+NO reaction rate, and they didn't consider this reaction when calculating the net production of OH from HONO. The authors used an unjustified circular assumption that OH loss in the OH+NO reaction at the ground will be recycled back to OH at a higher altitude without any valid calculation of HONO lifetime vs transport time from the ground to the summit. The authors claim that they calculated HOx budget, although they only calculated gross HONO photolysis and O3 photolysis. (primary sources of OH only). I suggest the authors limit their discussions to HONO sources and sinks, and that they should account for NO+OH reaction in calculating HONOpss or assume several OH values around those published earlier to calculate their uncertainties. Otherwise, the manuscript is publishable after addressing these comments.

Response: Thanks for your efforts and comments, which help to improve our manuscript. Please see the point-to-point response below (**Comments in Black; Response in Blue; Changes in Red**).

**Specific comments:**

Page 15, Line 320: The authors didn't justify the use of OH-$j(O^1D)$ correlation from previous publications to calculate OH in this study. Although some studies showed a good correlation, it still may not be a good proxy for OH given the large variation in the obtained slops. The authors use a circular argument that OH is not important since NO+OH is not important, to justify the uncertainty associated with their approach. At which OH levels does the NO+OH reaction accounts for 8%?

Maybe, it is safer to either simulate OH using a box model or use a range of OH levels around those reported previously by Kanaya et al. (2009) to show that it is not important, as they claim This is a major issue that the authors need to address before continuing with their calculations of unknown HONO sources.

The authors used several assumptions to calculate the contribution of different HONO sources to measured HONO levels. Most importantly is the photolysis of pNO3, for which the authors used a range of enhancement factors (EF) that ranges from 1 to ~15.6, accounting for 0.6 to 9.6%, depending on EF, leaving ~93% of HONO unknown sources unknown. I think. A major uncertainty here is related to HONOpss, which the authors didn't sufficiently address, which affects the unknown fraction HONO.

Response: We agree that the estimated OH could result in some uncertainties in calculation on unknown HONO sources and net OH production. We added OH sensitivity tests and found very small impacts on $P_{unknown}$. Figure S6 and the below texts are added in the manuscript.

The estimated OH could lead to some uncertainties. Hence, we added OH sensitivity tests to

reinforce our analysis and conclusion. The used OH, the corresponding HONO$_{pss}$, P$_{un}$ and results from the sensitivity tests were also shown in Figure S6. The estimated OH level was lower than that measured during the MTX campaign (Kanaya et al., 2013). This is mainly caused by lower J(O$^1$D) resulting from frequent cloudy weather during the present study period. For instance, the average RH during this campaign was 96%, which is much higher than that during the MTX campaign (67%). The variation of OH levels indeed remarkably impact HONO$_{pss}$. However, HONO$_{pss}$ (5-15 pptv level) is still 1-2 orders of magnitude lower than the observed HONO (50-200 pptv level), leading to a negligible impact of variable OH and HONO$_{pss}$ levels on P$_{un}$.

[Figure]

Figure S6: Estimated OH concentrations (red line) used in this study and corresponding HONO$_{pss}$ and P$_{un}$ (red lines). Black lines represent OH level reduced by 30% and corresponding HONO$_{pss}$ and P$_{un}$. Blue lines represent OH level enlarged by 30% and corresponding HONO$_{pss}$ and P$_{un}$.

Page 18, lines 404-412: The authors' argument of OH recycling via HONO photolysis as the source of OH at higher altitude is not justified and is flawed. The authors didn't provide information about the HONO lifetime vs the transport time to this altitude. I think this whole paragraph should be just deleted.

Response: The maximum of average diurnal J(HONO) is $8.0 \times 10^{-4}$ s$^{-1}$ (Figure 9), corresponding to a minimum HONO lifetime of about 21 min against photolysis, longer than the estimated transport time of 7-17.5 min. As shown in Figure 9, the remaining proportion of HONO after a period of transport from the ground to the summit levels is about 50-80% at noontime. α could be even larger because the calculation only considers HONO loss, whereas HONO production during the transport along the mountain slope was not taken into consideration. Then whether the transport of HONO could constitute an OH transport path depends on the amount of OH consumption to produce HONO through NO+OH at the foot station.

At the foot station, NO+OH contributed 15% of daytime HONO formation and photo-enhanced NO$_2$ uptake on the ground surface dominated the rest as reported in the companion ACP paper (Xue et al., 2021). Besides, hydrogen peroxide (H$_2$O$_2$), an important OH reservoir, could also be transported from the ground to the summit levels as reported in our recent study (Ye et al., 2021). At the ground level, H$_2$O$_2$ was mainly produced by HO$_2$+HO$_2$ (Ye et al., 2021). Hence, it could be preliminarily inferred that radicals (i.e., OH and HO$_2$) could be transported through their precursors/reservoirs (like HONO and H$_2$O$_2$) with lifetimes longer than themselves.

We improved the texts as:

Radicals, including OH and HO$_2$, are not expected to be transported far due to their short enough lifetimes (<1 s). However, 15% of daytime HONO was formed at the ground level through NO + OH as reported in the companion ACP paper (Xue et al., 2021), and part of OH consumed at the ground level would be released at the summit level through HONO photolysis. This could be supported by our recent finding that hydrogen peroxide (H$_2$O$_2$), an important OH reservoir, could be transported from the ground to the summit levels (Ye et al., 2021). At the ground level, H$_2$O$_2$ was mainly produced by HO$_2$+HO$_2$ (Ye et al., 2021). Hence, it could be preliminarily inferred that radicals (i.e., OH and HO$_2$) could be transported through their precursors/reservoirs (like HONO and H$_2$O$_2$) with lifetimes longer than themselves. Furthermore, the enhanced vertical air mass exchange could also lead to fast transport of other pollutants (PM$_{2.5}$, O$_3$, CO, SO$_2$, etc.) from the ground to the summit levels, which will significantly impact the atmospheric composition as well as its chemistry in the upper boundary layer or the residual layer. The discussion and implications in this study are instructive for further laboratory or model studies.

Page 18, line 414: provide a reference…
Response: A reference was added.
(Jiang et al., 2020)

Page 19, lines 418-420: This long sentence is not clear at all….either provide all relevant information or leave it for the accompanying paper. Otherwise, HONO net photolysis should be used to calculate HONO relative contribution to OH primary sources.
Page 19, lines 428-438: Again, this is all irrelevant if HONO net contribution is not calculated.
Response to both comments: The whole of Section 3.6 was improved as we replaced gross OH production from HONO photolysis with its net OH production (Figure 11). We also calculated the contribution P(OH)$_{HONO\_net}$ to P(OH)$_{sum}$ (Figure S8).
The improved figures and Section 3.6 are as follows:

[revised manuscript text omitted]

Page 19, lines 425-445: replace HOx with OH since you HONO and O3 photolysis are sources of OH only, not HO2.

Response: Done.

---

## Author Comment (AC2)

**Response to RC2**

This paper presents comprehensive field campaign which was performed in summer at the foot (150 m a.s.l) and the summit (1534 m a.s.l) of Mt. Tai (Shandong province, China). The author performed the analysis of HONO budget and found strong unknown HONO sources. Constraints on the kinetics of aerosol-derived HONO sources were discussed and their contribution to HONO formation were negligible. The vertical transport form the ground to the summit levels and heterogeneous conversion of $NO_2$ was proposed to support the remaining majority of unknown HONO sources. The subject is suitable for publication in ACP and I would recommend the paper is accepted after the author have addressed the following concerns.

Response: Thanks for your efforts and comments, which help to improve our manuscript. Please see the point-to-point response below (**Comments in Black; Response in Blue; Changes in Red**).

Specific comments:

Instrumentation: Low levels of HONO were measured by LOPAP technique with detection limit of 1.5 pptv at the summit of Mt. Tai in summer 2018. The QA and/or QC for LOPAP instrument should be stated to guarantee data quality.

Response: In Section 2.2 we added more information about the operation of the LOPAP instrument. At the summit station, a temperature-controlled measurement container was used to house all the instruments. The external sampling unit of LOPAP was installed on the top of the container, about 2.5 m above the ground surface. Zero air (ultrapure $N_2$) measurements were conducted 2 or 3 times per day. Liquid calibration with diluted standard nitrite solution (Sigma-Aldrich) was conducted every week. Both zero air measurements and liquid calibration were conducted after changing any solution, cleaning the instrument, or replacing any component of the instrument (the air pump was broken on 21st July and replaced by a new one on 25th July). The precision of the instrument determined from $2\sigma$ noise of the calibration was 1%. An accuracy of 7% was determined by error propagation including all known uncertainties, i.e., the concentration of the calibration standard ($\pm 3$-4%) and the liquid ($\pm 1\%$) and gas flow ($\pm 2\%$) rates. Known artificial HONO formation on inlet surfaces (e.g., Zhou et al., 2002) were minimized by using the external sampling unit, with only a 3 cm sunlight-shielded glass inlet to the ambient atmosphere. Other interferences were considered of minor importance, as they were corrected for by the two-channel concept of the instrument. In addition, excellent agreement between LOPAP and DOAS techniques was observed under complex conditions in a smog chamber and in the ambient atmosphere (Kleffmann et al., 2006).

Anthropogenic emissions: The author stated that low $NO_x/NO_y$ of $0.43 \pm 0.28$ indicated aged air masses and small impact of anthropogenic emissions. However, NO and $NO_x$ were measured simultaneously at the summit station. Why did not the author utilize $NO/NO_x$ to evaluate the influence of nearby anthropogenic emissions. Moreover, the rapid increase in pollutants (HONO, NO, $NO_2$, $NO_y$, CO, $PM_{2.5}$) was observed on 29 July. Low NO concentrations (1-2 ppbv) were observed at high $O_3$ levels (~50 ppbv), which should originate from local emissions. However, the author stated the high HONO levels could come from the heterogeneous conversion. The author should reexamine the data and explore the sources of increased pollutants.

Response: As demonstrated in Section 2.1, potential anthropogenic emissions could happen around the Southern Heavenly Gate, the Bixia Temple, and the Jade Emperor Peak. All of the three places

are within 1 km west of our station. If emissions originated from those regions, sharp peaks would be observed and the $NO/NO_x$ ratio should be near to that of fresh plumes.

However, this event lasted about 1.5 hours (5:20-6:50), much longer than the duration of the fresh plumes observed at the foot station. Besides, during this event, air mass originated from the south (Figure 2), the polluted urban region rather than the direction of the potential sources at the summit level. Furthermore, the $NO/NO_x$ ratio in this plume is 0.21, lower than fresh combustion plume with a $NO/NO_x$ ratio of ~0.9 or even higher (Carslaw and Beevers, 2005; He et al., 2020; Kurtenbach et al., 2012; Wild et al., 2017). This is also lower than the fresh plumes observed at the foot station with an average $NO/NO_x$ ratio of $0.46\pm0.19$ at high $O_3$ levels (Xue et al., 2021).

Therefore, we could conclude that the observed plume should originate from transport from the foot urban region rather than nearby emissions at the summit.

We then improved related texts as:

During this event, air mass originated from the south (Figure 2), the polluted urban region (Figure S1E) rather than the direction of the potential sources at the summit level. This event lasted about 1.5 hours (5:20-6:50), much longer than the duration of the typical fresh plumes observed at the foot station. Furthermore, the $NO/NO_x$ ratio of this plume was 0.21, lower than the direct $NO/NO_x$ emission ratio of ~0.9 (Carslaw and Beevers, 2005; He et al., 2020; Kurtenbach et al., 2012; Wild et al., 2017). This is also lower than that of the fresh plumes observed at the high-$O_3$ foot station with an average $NO/NO_x$ ratio of $0.46\pm0.19$ (Xue et al., 2021). Therefore, we could conclude that the observed plume should originate from the foot urban region rather than nearby emissions at the summit. The $\Delta HONO/\Delta NO_x$ within this plume was 8%, much larger than that inferred from direct emissions (typically inferred as less than 1%). The ratio could be enhanced by: 1) night-time $NO_2$-to-HONO conversion at the ground level where the air mass was already aged before being transported to the summit level, 2) in-plume $NO_2$-to-HONO conversion along the mountain slope (rock and vegetation surfaces, etc.), and 3) in-plume $NO_2$-to-HONO conversion on particle surfaces as both the boundary layer height (BLH) elevation and the valley breeze are initialized after sunrise.

Figure 5: The data of HONO and $J(NO_2)$ for summit and foot station were measured at different periods. Whether it is appropriate to exhibit the data at different periods together in the figure? The measured data at different periods were different. Is such comparison meaningful?

Response: Measurements at the foot and the summit stations represent typical average diurnal variations for ground surface or summit measurements, respectively. Similar pattern of variations have also been reported by many previous studies including ground surface measurements (Alicke et al., 2002, 2003; Gu et al., 2020; Hendrick et al., 2014; Kleffmann et al., 2005; Platt et al., 1980; Su et al., 2008) and summit measurements (Jiang et al., 2020; Kleffmann et al., 2002; Kleffmann and Wiesen, 2008). Besides, our measurements at the two stations were conducted during two consecutive periods in summer 2018. To confirm our argument, we also compared pollutants at the ground and summit stations during the same period, such as $PM_{2.5}$, CO, $O_3$, and $SO_2$ (Figure 7 in the manuscript) discussed in Section 3.2.2.3 of our manuscript.

Hence, the comparison could allow potential insights into the link between atmospheric chemistry at the ground surface and summit levels.

Page 13, line 285-290: The author stated that south wind could enhance the upslope valley breeze wind because higher wind speeds (>5 m s$^{-1}$) were observed at the summit station than at the foot of

the mountain (> 2 m s$^{-1}$). However, the wind speeds are generally higher at the summit station, which requires detailed explanation by the author.

Response: The fact that the south wind could enhance the upslope valley breeze wind is not because of higher wind speed at the summit level. It's because the urban site (150 m a.s.l.) is south of the summit station (1534 m a.s.l.).

The reported upslope valley breeze wind speed was about 2-5 m s$^{-1}$. With consideration of south wind at the ground level (>2 m s$^{-1}$), the integrated wind speed along the mountain slope could be 4-7 m s$^{-1}$. Alternatively, with consideration of south wind at the ground level (>5 m s$^{-1}$), the integrated wind speed along the mountain slope could be 7-10 m s$^{-1}$. Therefore, we used the wind speed range of 4-10 m s$^{-1}$ to consider all the possible situations.

Related texts are improved as:

The upslope valley breeze wind could transport polluted air mass from the foot to the summit levels. This process could be accelerated by the dominant south wind (Figure 8) as the urban site (150 m a.s.l.) is south of the summit station (1534 m a.s.l.). The mean south winds measured at the ground and summit stations are >2 and >5 m s$^{-1}$, respectively. Then the integrated wind speed along the mountain slope should be 4 – 10 m s$^{-1}$, and the calculated $t_{transport}$ will be reduced to 7 – 17.5 min.

Page 16, line 347-348: "Note that the uncertainty of …". I don't quite understand this sentence. Section 3.6 stated the contribution of photolysis of HONO and O$_3$ to OH. Please give the explanation.

Response: We cited Figure 10 and added the below discussion on the contribution (3%) of P(HONO)$_a$ to HONO formation. It has been revised as:

Note that the uncertainty of S$_a$ is not expected to cause a significant uncertainty on HONO budget analysis as P(HONO)$_a$ was not the dominant source (Figure 10 and see the below discussion on P(HONO)$_a$ contribution).

The author calculated the enhanced uptake coefficient of NO$_2$ on the aerosol surfaces. The dark uptake of NO$_2$ on the aerosol surface could be considered to evaluate the influence of heterogeneous reaction on the aerosol surfaces since the dark uptake coefficient of NO$_2$ were mostly investigated.

Response: From the correlation analysis (Table 4), we found poor correlations (r = 0.17 or 0.64) between P$_{un}$ and NO$_2$*S$_a$ or NO$_2$*S$_a$*J(NO$_2$), suggesting minor roles of dark and photo-enhanced NO$_2$ uptake on the aerosol surface in the HONO formation.

Besides, with $\gamma_a$ = 2×10$^{-5}$, photo-enhanced P(HONO)$_a$ could only explain 3% of P$_{un}$. For dark NO$_2$ uptake, $\gamma_{a\_dark}$ is generally at a level of 10$^{-6}$ (George et al., 2005; Han et al., 2017; Stemmler et al., 2006, 2007), implying that P(HONO)$_{a\_dark}$ is much lower than P(HONO)$_a$. Therefore, we didn't consider dark NO$_2$ uptake on the aerosol surface.

The following sentence was added in Section 3.4:

Note that dark NO$_2$ uptake on the aerosol surface was not considered due to a much lower uptake coefficient generally at a level of 10$^{-6}$ (George et al., 2005; Han et al., 2017; Stemmler et al., 2006, 2007).

Page 19, Section 3.6: The author only calculated the contribution of the photolysis of HONO and O$_3$ to OH and not HO$_x$. The HO$_x$ should be replaced by OH.

Response: Done.

Page 21, line 470: What dose $\lambda_a$ stand for? It is $\gamma_a$?

Response: It has been changed to $\gamma_a$.

[revised manuscript text omitted]

---

## Author Response (AR2)

**Comments to the author**:

Dear Authors,

This time, the revised manuscript as well as the Authors' responses were evaluated by the Reviewer #1 and myself. Though several parts were improved, still major revision of the manuscript is needed. See further comments from the reviewer #1 and my points below.

Response: Thanks for the comments from the editor and reviewer #1. Please see the point-to-point response below (Comments in Black; Response in Blue; Changes in Red). Line numbers mentioned in this reply refer to our revised version with changes tracked.

1. About estimated OH: the authors' description was improved but the main point from the reviewer #1 is still not addressed. It is read from line 347 (change-track version) that relationship between OH and J(O1D) was taken from (Kanaya et al., 2009) for the summit of Mt. Tai for June 2006 but it is not found there. Is this relationship from other study? Clarification is necessary. Irrespective of the answer to this question, OH concentration levels need to be estimated "with" the measured HONO concentration in this case for consistency with other parts of this study. One possibility would be to linearly scale the OH levels with the enhancement of the primary production due to the HONO photolysis, while a full box model simulation (i.e., deriving HO2 levels and quantifying OH production from the HO2 + NO reaction) is recommended. I believe that the P_un cannot be explained by this revised treatment but consistency and quantitativeness are necessary. Also please note that OH from Kanaya et al. (2009, 2013) was not MEASURED but "estimated" from a box model constrained with observations (lines 351-352).

Response: Regarding the relationship of OH and $J(O^1D)$, it was indeed taken from Kanaya et al. (2009), in which radical chemistry was modeled and discussed. From the averaged diurnal variations of $J(O^1D)$ (Fig. 1) and the modeled OH (Fig. 2) high correlations ($R^2>0.9$) were found between them, indicating very similar diurnal variations of OH to that of $J(O^1D)$. However, this does not yield high correlations between the time series of modeled OH and that of $J(O^1D)$, as the editor commented that correlation between OH and $J(O^1D)$ was not found during the MTX 2006 campaign. Therefore, we can conclude that the present OH estimation method can predict the diurnal variations of OH but not its absolute levels due to 1) high correlations between OH and $J(O^1D)$ was found in their average diurnal variations but not in their time series, and 2) HONO was not constrained in the box model simulations for the MTX 2006 campaign.

One way to consider HONO impact is to discuss the uncertainties caused by HONO when using the correlation between OH and $J(O^1D)$ during the MTX 2006 campaign. Assuming that the impact of HONO on OH levels is determined by its contribution to primary OH production at both the foot and the summit stations, we can preliminarily deduce the present OH estimation uncertainties caused by the lack of HONO chemistry. At the foot station, HONO photolysis made a contribution of 64% to primary OH production (see Section 3.6). If the box model was not constrained by the measured HONO, OH would be underestimated by 25% (see the companion paper, Xue et al.

(2022)). At the summit station, HONO contributed 18% of the primary OH production (see Section 3.6). Therefore, OH underestimation due to HONO chemistry at the summit station should be roughly around 7% (25% divided by 64% and multiplied by 18%).

Regarding the suggestion on box model simulations, because VOCs data were not available, we can't run box models with VOCs and HONO constrained to estimate OH levels.

Considering that Mt. Tai is surrounded by polluted regions, empirical formulas between OH and $J(O^1D)$ from the ground or other mountain measurements, may not be reasonable here. Hence, to cover the uncertainties caused by the above issues, we made sensitivity tests on OH levels by increasing or reducing its level by 30%.

As the editor/reviewer also agrees with our argument that the OH level doesn't significantly impact $P_{un}$ as well as further analysis based on the obtained $P_{un}$, we will improve the text about how the present OH was estimated and point out its uncertainties as well.

Changes in the manuscript (L346-373):

OH measurements were not available during this campaign. One popular used method to estimate OH is based on the correlation between OH and solar ultraviolet radiation (e.g., $J(O^1D)$) (Rohrer and Berresheim, 2006). Considering that Mt. Tai is surrounded by polluted regions, empirical formulas between OH and $J(O^1D)$ from the ground or other mountain measurements, may not be reasonable here. In June 2006 Kanaya et al. conducted a comprehensive field campaign at the summit of Mt.Tai (Kanaya et al., 2009). OH levels and sources were studied by a box model. From the average diurnal variations of the modeled OH and $J(O^1D)$, a significant correlation ($R^2>0.9$) between them was found, which was used here to estimate OH concentrations. This method could lead to some uncertainties in OH levels due to: 1) high correlations between OH and $J(O^1D)$ was found in their average diurnal variations but may not be in their time series, and 2) HONO was not constrained in the box model simulations so that OH could be underestimated for the MTX 2006 campaign.

One way to consider HONO impact is to discuss the OH uncertainties caused by the lack of HONO chemistry. Assuming that the impact of HONO on OH levels is determined by its contribution to primary OH production, we can preliminarily deduce the OH uncertainties at the summit station caused by the lack of HONO chemistry based on measurements and model simulations for the foot station. At the foot station, HONO photolysis made a contribution of 64% to primary OH production (see Section 3.6). If the box model was not constrained by the measured HONO, OH would be underestimated by 25% (see the companion paper, Xue et al. (2022)). At the summit station, HONO contributed 18% of the primary OH production (see Section 3.6). Therefore, OH underestimation due to the lack of HONO chemistry at the summit station should be roughly around 7% or so.

Hence, to cover the uncertainties caused by the above issues, we added OH sensitivity tests, reducing or increasing the OH level by 30%, to quantify the impact of the OH uncertainties in our further analysis and conclusion. The used OH, the corresponding $HONO_{pss}$, $P_{un}$ and results from the sensitivity tests were also shown in Figure S6. The

estimated OH level was lower than that modeled during the MTX campaign (Kanaya et al., 2009, 2013). This is mainly caused by lower $J(O^1D)$ resulting from frequent cloudy weather during the present study period. For instance, the average RH during this campaign was 96%, which is much higher than that during the MTX campaign (67%). Reducing or enlarging OH levels by 30% indeed remarkably impact $HONO_{pss}$. However, $HONO_{pss}$ (5-15 pptv level) is still 1-2 orders of magnitude lower than the observed HONO (50-200 pptv level), leading to a small impact of variable OH and $HONO_{pss}$ levels on $P_{un}$. Hence, we highlight the uncertainties in OH levels estimated by the current method, but its impact on following HONO budget analysis should be small as discussed above.

2. The concept that HONO is transported with the upslope wind is interesting, but is valid if only after verifying that dispersion/diffusion during the transport is also slower than the time constant. This dispersion/diffusion term could be estimated from longer-lived species (e.g., CO or SO2). Such discussion is needed before conclusion in lines 329-333.
Response: Thanks for the suggestion. Discussion about dilution and dispersion is a very good point to have more insights into the concept proposed in this study. We compared CO levels (thanks for the guidance) at the foot and the summit stations to preliminarily deduce the dilution effect and revised the related discussion.
(L320-330)
Figure 9 shows the calculated $\alpha$ with $t_{transport}$ = 7 or 17.5 min during the daytime. It is apparent that $\alpha$ is larger than 43% with $t_{transport}$ = 17.5 min and larger than 72% with $t_{transport}$ = 7 min, providing a theoretical basis for the potential role of vertical HONO transport from the ground to the summit levels. The calculations don't consider the atmospheric dilution or dispersion during the transport, which may reduce $\alpha$. This effect could be roughly quantified by comparing levels of long lifetime species (e.g., CO) at the foot and the summit stations. Hourly CO averages at noon are 493 and 379 ppbv measured at the foot and the summit stations, respectively (Figure 5). This preliminarily indicates a dilution factor of 1.3. The dilution process may also similarly affect HONO, i.e., $\alpha$ is expected to be reduced by a factor of 1.3, leading to $\alpha$ values of >55% and >33% with $t_{transport}$ = 7 or 17.5 min, respectively. The above calculation only included the daytime HONO sink through photolysis and atmospheric dilution, but the sources, such as NO + OH and heterogeneous $NO_2$ reactions, were not considered, and hence, the calculated $\alpha$ represents a lower limit.

3. OH generation from H2O2 photolysis would be of minor importance (4% of OH production term from Kanaya et al, 2009, or even less when considering HONO photolysis) and thus the description in lines 445-450 is not readily supported. The authors need a quantitative counter statement about the contribution when the statement is to be kept.
Response: Thanks for the information about the contribution of $H_2O_2$ photolysis to OH. We agree that the contribution indeed is expected to be less when HONO was considered.

Besides, we estimated the OH production from $H_2O_2$ photolysis ($P(OH)_{H2O2}$). With noontime $H_2O_2$ of 1.4 ppbv measured in summer 2019 (Ye et al., 2021) and maximum photolysis frequency of $7.0 \times 10^{-6}$ $s^{-1}$, the calculated $P(OH)_{H2O2}$ is 0.04 ppbv $h^{-1}$, which is much lower than $P(OH)_{HONO\_net}$ or $P(OH)_{O3}$ of 0.2 and 1.0 ppbv $h^{-1}$, respectively. Therefore, we deleted the discussion about the role of $H_2O_2$ in OH production.

4. The righthand side of the equation 8 must be [HONO]xJ(HONO) - k_1[NO][OH] - k_2[HONO][OH]. Here the OH levels are again necessary to take into account (similar to the point 1 above; the OH level should be determined with the impact from HONO photolysis). Correction is necessary when the calculations were not made adequately.
Response: The typing mistake in Equation 8 was corrected. Note that during our calculation in Excel, the right formulas were used.
Regarding the comment on OH level, please see the above response to comment 1.

**References**
Kanaya, Y., Pochanart, P., Liu, Y., Li, J., Tanimoto, H., Kato, S., Suthawaree, J., Inomata, S., Taketani, F., Okuzawa, K., Kawamura, K., Akimoto, H. and Wang, Z. F.: Rates and regimes of photochemical ozone production over Central East China in June 2006: A box model analysis using comprehensive measurements of ozone precursors, Atmos. Chem. Phys., 9(20), 7711–7723, doi:10.5194/acp-9-7711-2009, 2009.
Kanaya, Y., Akimoto, H., Wang, Z. F., Pochanart, P., Kawamura, K., Liu, Y., Li, J., Komazaki, Y., Irie, H., Pan, X. L., Taketani, F., Yamaji, K., Tanimoto, H., Inomata, S., Kato, S., Suthawaree, J., Okuzawa, K., Wang, G., Aggarwal, S. G., Fu, P. Q., Wang, T., Gao, J., Wang, Y. and Zhuang, G.: Overview of the Mount Tai Experiment (MTX2006) in central East China in June 2006: Studies of significant regional air pollution, Atmos. Chem. Phys., 13(16), 8265–8283, doi:10.5194/acp-13-8265-2013, 2013.
Rohrer, F. and Berresheim, H.: Strong correlation between levels of tropospheric hydroxyl radicals and solar ultraviolet radiation, Nature, 442(7099), 184–187, doi:10.1038/nature04924, 2006.
Xue, C., Ye, C., Kleffmann, J., Zhang, W., He, X., Liu, P., Zhang, C., Zhao, X., Liu, C., Ma, Z., Liu, J., Wang, J., Lu, K., Catoire, V., Mellouki, A. and Mu, Y.: Atmospheric measurements at Mt. Tai – Part II: HONO budget and radical ($RO_x$ + $NO_3$) chemistry in the lower boundary layer, Atmos. Chem. Phys., 22(2), 1035–1057, doi:10.5194/acp-22-1035-2022, 2022.
Ye, C., Xue, C., Zhang, C., Ma, Z., Liu, P., Zhang, Y., Liu, C., Zhao, X., Zhang, W., He, X., Song, Y., Liu, J., Wang, W., Sui, B., Cui, R., Yang, X., Mei, R., Chen, J. and Mu, Y.: Atmospheric Hydrogen Peroxide ($H_2O_2$) at the Foot and Summit of Mt. Tai: Variations, Sources and Sinks, and Implications for Ozone Formation Chemistry, J. Geophys. Res. Atmos., 126(15), 1–14, doi:10.1029/2020JD033975, 2021.

---

## Author Response (AR3)

**Comments to the author**:

Dear authors,

I see the revised manuscript has been improved. However, discussion on the dilution factor using CO has a flaw. Background CO concentration level must be subtracted before the analysis. Assuming your lowest CO concentration (167 ppbv) as background, hourly CO averages around noon are 493 and 379 ppbv should result in a much higher dilution factor, around 1.54.

Else the revision is successful, though English needs to be improved.

Yugo Kanaya, ACP Editor

Response:

Dear Prof. Kanaya,

Thanks for the comments. Please see the response below (Comments in Black; Response in Blue; Changes in Red). Line numbers mentioned in this reply refer to our revised version with changes tracked.

The English of this manuscript was improved. In addition, we recalculated the dilution factor. Noontime CO at the summit station was subtracted by the background CO level that was assumed as the CO minima of the summit measurements. We didn't do that for the foot CO level because the plume was diluted by the air along the slope of the mountain and at the summit level.

Changes in L324-327

The measured hourly CO averages at noon are 493 and 379 ppbv at the foot and the summit stations, respectively (Figure 5). Taking the minima of CO measurements as the background CO level (167 ppbv), we can obtain a dilution factor of 2.3. The dilution process may also similarly affect HONO, i.e., $\alpha$ is expected to be reduced by a factor of 2.3, leading to $\alpha$ values of >31% and >19% with $t_{transport}$ = 7 or 17.5 min, respectively.